# RalGPS2 Interacts with Akt and PDK1 Promoting Tunneling Nanotubes Formation in Bladder Cancer and Kidney Cells Microenvironment

**DOI:** 10.3390/cancers13246330

**Published:** 2021-12-16

**Authors:** Alessia D’Aloia, Edoardo Arrigoni, Barbara Costa, Giovanna Berruti, Enzo Martegani, Elena Sacco, Michela Ceriani

**Affiliations:** 1Department of Biotechnology and Biosciences, University of Milano-Bicocca, Piazza della Scienza 2, 20126 Milan, Italy; alessia.daloia@unimib.it (A.D.); edoardo.arrigoni@unimib.it (E.A.); barbara.costa@unimib.it (B.C.); enzo.martegani@unimib.it (E.M.); elena.sacco@unimib.it (E.S.); 2Department of Biosciences, University of Milan, Via Celoria 26, 20133 Milan, Italy; giovanna.berruti@unimi.it; 3SYSBIO-ISBE-IT-Candidate National Node of Italy for ISBE, Research Infrastructure for Systems Biology Europe, 20126 Milan, Italy; 4Milan Center for Neuroscience (NeuroMI), University of Milano-Bicocca, Piazza dell’Ateneo Nuovo 1, 20126 Milano, Italy

**Keywords:** tunneling nanotubes, bladder cancer, kidney, microenvironment, stress condition, RalGPS2, Akt, PDK1

## Abstract

**Simple Summary:**

Cell-to-cell communication in the tumor microenvironment is a crucial process to orchestrate the different components of the tumoral infrastructure. Among the mechanisms of cellular interplay in cancer cells, tunneling nanotubes (TNTs) are dynamic connections that play an important role. The mechanism of the formation of TNTs among cells and the molecules involved in the process remain to be elucidated. In this study, we analyze several bladder cancer cell lines, representative of tumors at different stages and grades. We demonstrate that TNTs are formed only by mid or high-stage cell lines that show muscle-invasive properties and that they actively transport mitochondria and proteins. The formation of TNTs is triggered by stressful conditions and starts with the assembly of a specific multimolecular complex. In this study, we characterize some of the protein components of the TNTs complex, as they are potential novel molecular targets for future therapies aimed at counteracting tumor progression.

**Abstract:**

RalGPS2 is a Ras-independent Guanine Nucleotide Exchange Factor for RalA GTPase that is involved in several cellular processes, including cytoskeletal organization. Previously, we demonstrated that RalGPS2 also plays a role in the formation of tunneling nanotubes (TNTs) in bladder cancer 5637 cells. In particular, TNTs are a novel mechanism of cell–cell communication in the tumor microenvironment, playing a central role in cancer progression and metastasis formation. However, the molecular mechanisms involved in TNTs formation still need to be fully elucidated. Here we demonstrate that mid and high-stage bladder cancer cell lines have functional TNTs, which can transfer mitochondria. Moreover, using confocal fluorescence time-lapse microscopy, we show in 5637 cells that TNTs mediate the trafficking of RalA protein and transmembrane MHC class III protein leukocyte-specific transcript 1 (LST1). Furthermore, we show that RalGPS2 is essential for nanotubes generation, and stress conditions boost its expression both in 5637 and HEK293 cell lines. Finally, we prove that RalGPS2 interacts with Akt and PDK1, in addition to LST1 and RalA, leading to the formation of a complex that promotes nanotubes formation. In conclusion, our findings suggest that in the tumor microenvironment, RalGPS2 orchestrates the assembly of multimolecular complexes that drive the formation of TNTs.

## 1. Introduction

According to Global Cancer Statistics 2020 [1], bladder and kidney cancers together accounted for over 1 million new cases and about 400.000 new deaths worldwide in 2020.

Most new cases of cancer are diagnosed when tumors are localized and confined within the organ affected, with a relatively high 5-year survival rate (~95%). However, many patients experience relapse within 5 years, some of them progressing to invasive disease with a significant drop in life expectancy (5-year relative survival rate of 13% and 6% for metastatic kidney and bladder cancer, respectively) [2,3,4].

In particular, the cost of therapies for bladder cancer is the highest, compared to other cancers in the United States [5,6] and Europe, and has a huge impact on public health costs [7].

Novel safe and effective therapeutic approaches for the treatment of bladder cancer are urgently needed. To this end, it is necessary to further investigate the molecular mechanisms underlying tumor progression, invasion and metastasis formation and identify new pharmacological targets.

In the tumor microenvironment, cell-to-cell communication and signaling mechanisms play a pivotal role, enabling the exchange between healthy and cancer cells of signaling molecules, metabolites and organelles, such as mitochondria. Such exchange mechanisms enable cancer cells to affect the healthy ones, causing the latter to develop tumor-supportive traits [8].

Tunneling nanotubes (TNTs) are highly-dynamic membrane protrusions that enable cells to directly communicate with each other over long distances (>120 μm) [9,10]. Indeed, TNTs are long-range cytoplasmic channels that enable the exchange between cells of cargoes containing organelles, molecules, proteins, pathogens, miRNAs and ions [11,12,13,14,15,16,17,18]. Lou et al. [19] first demonstrated the presence of TNTs in tumors in human malignant pleural mesothelioma. Over the years, it has become clear that these TNTs are involved in intercellular communication processes during early development, cell migration, stem cell-mediated homeostasis and regeneration, as well as advanced neurodegeneration, cancer progression and metastasis development [16]. The formation of TNTs occurs in several cell types, including neuronal cells, epithelial cells and most cells of the immune system [20,21,22,23,24,25,26]. TNTs are also present in tumors, such as urothelial carcinoma, cervix carcinoma, breast and colon cancer, glioblastoma, leukemia, mesothelioma [10,19], ovarian cancer [27], osteosarcoma [28] and pancreatic cancer [29]. TNTs are not branched and are suspended above the matrix [26], they can fall into two different types, each of which can be present simultaneously in the same cell type [30]: (i) type I, essentially made of actin, is formed by cells within their surroundings and ii) type II, made of tubulin and cytokeratin filaments, which is formed by the detachment of two cells that are already connected [31,32]. Type I are short (100–200 nm), thin (≤0.7 µm) and mostly dynamic structures, while type II are long (1 µm), thick (≥0.7 µm) and stable [33]. TNTs can be closed or open-ended; the latter case allowing the cytoplasm of the two connected cells to mix [9,34]. However, the molecular mechanisms responsible for the formation of TNTs have not been fully elucidated yet. Gousset et al. have demonstrated that TNTs formation is dependent on the unconventional motor protein myosin X, which interacts with several transmembrane proteins [35]. Specifically, Gousset et al. showed that the transmembrane MHC class III protein leukocyte-specific transcript 1 (LST1) recruits to the plasma membrane the actin cross-linking protein filamin and interacts with myosin, myoferlin and M-Sec [36]. The latter, M-Sec, also known as TNFαIP2 (tumor necrosis factorα-induced protein), together with the RalA small GTPase, promotes the assembly of a multi-protein complex (exocyst complex), which can induce the formation of functional TNTs [37]. TNTs formation in different cell types and models depends on different signaling mechanisms [38]. For instance, it has been shown that elevated levels of p53 in astrocytes are essential for TNTs formation, though this is not the case in other cell types, such as pheochromocytoma PC12 cells [33,39]. Furthermore, in 2018, our group demonstrated, in 5637 bladder cancer cell lines, the role of RalGPS2 (Ral GEF with PH domain and SH3-binding motif 2), a Ras-independent guanine exchange factor (GEF) for the GTPase RalA, in TNTs formation under low serum condition [40]. Although in normal serum conditions, RalA GTPase is predominantly activated by Ral guanine nucleotide dissociation stimulator (RalGDS), Ras-dependent GEFs activated by Ras GTPase [41,42,43], once activated, RalA interacts with Sec5, promoting the assembly of the exocyst complex and regulating exocytosis and cell proliferation. Aside from this role, RalGPS2 also affects protein kinase B (Akt) activation, leading to its phosphorylation. Phospho-Akt (Akt P) regulates, in turn, TNTs formation through the mechanistic target of rapamycin (mTOR) pathways, triggering F-actin polymerization and promoting TNTs development [44].

In this work, we investigated TNTs formation in bladder cancer cell lines at different stages and grades, and we could demonstrate that cancer cells at mid or high -stages generate functional TNTs, which are able to exchange mitochondria. Moreover, in 5637 cell lines, RalA GTPase and LST1 are transported through TNTs. Finally, our data demonstrate, both in 5637 and HEK293 cell lines, that RalGPS2 GEF is crucial for TNTs development through the assembly of molecular machinery formed by Akt and PDK1, in addition to LST1 and RalA.

## 2. Materials and Methods

### 2.1. Plasmids and Constructs

The plasmid expressing the fusion protein GFP-RalGPS2 was described previously [45]. The pEGFP-C1 vector was purchased from Clontech (Clontech Laboratories Inc., Mountain View, CA, USA). The mCherry-RalA expression construct [46] was kindly provided by J. Camonis (CNRS Inserm Institut Curie, INSERM U830, Paris, France). The mCherry-LST1 expression construct [36] was kindly provided by C. Schiller (Department of Biology II, Ludwigs-Maximilians-Universität München, Großhadernerstr. 2, 82,152 Planegg-Martinsried, München, Germany).

### 2.2. Antibodies

Mouse anti-RalA antibodies (R23520) were from BD Transduction Laboratories (San Jose, CA, USA; dilution 1:1000). Anti-RalGPS2 rabbit antibodies have been described in a previous article (dilution 1:3000) [40,45]. Rabbit primary anti-GAPDH (#2118; dilution 1:3000), rabbit monoclonal Phospho-Akt (Ser473) (D9E; dilution 1:1000) and polyclonal anti-Akt (C67E7; dilution 1:1000) antibodies were purchased from Cell Signaling (Cell Signaling Technology, Danvers, MA, USA). The mouse monoclonal anti-Sec5 (F-7; dilution 1:200), anti-PDK1 (4A11F5; dilution 1:200), anti-Phospho-p53 (D-9; dilution 1:200) and TNFαIP2 (F6; dilution 1:1000) antibodies were obtained from Santa Cruz Biotechnology (Santa Cruz Biotechnology, Dallas, TX, USA). Anti-LST1 monoclonal antibodies (dilution 1:200) were previously described [36,47]. The secondary antibodies for western blot, peroxidase-conjugated sheep anti-mouse (515-035-003), donkey anti-rabbit (711-035-152) and anti-rat (712-036-150) polyclonal antibodies, were obtained from Jackson Ltd. (Jackson Ltd., Philadelphia, PA, USA; dilution 1:5000).

### 2.3. Cell Cultures

Grade 2 RT112 (stage pTa) and 5637 (stage not reported (nr)) bladder cancer cell lines were obtained from the American Type of Culture Collection (ATCC, Manassas, VA, USA). Grade 1-2 RT4 (stage pT1) and grade 3 HT1376 (stage ≥ pT2) bladder cancer cell lines were kindly provided by R. Vago (Urological Research Institute, division of experimental oncology, IRCCS, San Raffaele Hospital, Milan, Italy). Grade nr UMUC-3 (stage pT2–4) and grade 3 J82 (stage pT3) bladder cancer cell lines were purchased from Elabscience (Elabscience Biotechnology Inc., Houston, TX, USA). For bladder cancer cell lines classification, we referred to Zuiverloon et al. [48].

Cell lines were grown in RPMI-1640 medium (Merck Life Science, Darmstadt, Germany) supplemented with heat-inactivated 10% fetal bovine serum (FBS, Gibco-ThermoFisher, Waltham, MA, USA), 2 mM glutamine, 100 U/mL penicillin and 100 mg/mL streptomycin, at 37 °C in a humidified atmosphere of 5% CO_2_ (unless specified otherwise, cells were maintained under these conditions). Cells were passed using trypsin-ethylenediaminetetraacetic acid (EDTA). Live imaging was performed with the Operetta CLS™ High-Content Analysis System (PerkinElmer, Inc, Waltham, MA, USA). Unless otherwise indicated, cell assays were performed in experimental medium: Dulbecco’s modified Eagle’s medium (DMEM) *w/o* phenol red (Gibco™-Thermo Fisher Scientific), FBS 10%, 10 mM glucose, 2 mM glutamine, 100 U/mL penicillin and 100 mg/mL streptomycin (unless specified, all reagents were from Merck Life Science). HEK293 Phoenix, human embryonic kidney cell line, kindly provided by prof. Renata Zippel (University of Milan, Italy), was grown at 37 °C and 5% CO_2_ in DMEM (Euroclone, Pero, Italy) supplemented with 10% FBS, 2 mM glutamine, 100 U/mL penicillin and 100 mg/mL streptomycin (all Euroclone, Pero, Italy).

### 2.4. Tunneling Nanotubes Analysis in Bladder Cancer Cell Lines

RT4, RT112, 5637, HT1376, UMUC-3 and J82 cells were plated at a density of 1 × 10^4^ cells/well on CellCarrier Ultra 96-well Microplates (PerkinElmer, Inc, Waltham, MA, USA) in 100 µL of the experimental medium at 37 °C and 5% CO_2_. After 24 h, cell images were acquired using Operetta CLS™ (PerkinElmer, Inc, Waltham, MA, USA) equipped with 63× immersion objective in brightfield and Digital Phase Contrast (DPC) at 37 °C and 5% CO_2_. Cells were carefully examined for the presence of tunneling nanotubes. We considered as positive any cell with at least one TNT. For quantitative determination, the percentage of cells forming nanotubes was calculated. About 200 cells, derived from different view fields of a plate, for each experiment were analyzed. The analysis was performed at the magnification described above. Experiments were performed in triplicate. Data were analyzed using GraphPadv6.0 software (San Diego, CA, USA) using ANOVA followed by Tukey’s test for group comparison. *p* < 0.05 was considered statistically significant.

### 2.5. Mitochondrial Transfer via TNTs

5637, HT1376, UMUC-3 and J82 cells were plated at a density of 1 × 10^4^ cells/well on Cell Imaging 24-well Plates (Eppendorf, Hamburg, Germany) in 500 µL of the experimental medium at 37 °C and 5% CO_2_. After 24 h, cells were stained with 200 nM MitoTracker™ Green FM (Invitrogen™ Thermo Fisher Scientific, Waltham, MA, USA) in an FBS-free medium for 20 min at 37 °C and 5% CO_2_. To assess active mitochondria, cells were plated as previously described and, after 24 h, stained with 200 nM MitoTracker™ Green FM and 100nM Tetramethylrhodamine, Ethyl Ester, Perchlorate (TMRE, Invitrogen™ Thermo Fisher Scientific, Waltham, MA, USA), in FBS-free medium for 20 min at 37 °C and 5% CO_2_. Before image acquisition, the staining medium was replaced with the experimental medium. Cell images were acquired using Operetta CLS™ (PerkinElmer, Inc, Waltham, MA, USA) equipped with 63× immersion objective in brightfield, Digital Phase Contrast (DPC) and fluorescence to detect MitoTracker. The instrument was set at 37 °C and 5% CO_2_. Cells were carefully scored for the presence of mitochondria within TNTs. To perform quantitative determination, the percentage of cells able to exchange mitochondria via TNTs was calculated. About 200 cells for each experiment were assessed and data analyzed using GraphPadv6.0 software (San Diego, CA, USA) employing ANOVA followed by Tukey’s test for group comparison. *p* < 0.05 was considered statistically significant.

### 2.6. Transfection

Transient transfection of 5637 cells was performed with mCherry-RalA or mCherry-LST1 or GFP-RalGPS2 expression vectors, as previously described [40].

### 2.7. Time-Lapse Imaging

5637 cells were seeded at the concentration of 1 × 10^4^ cells/well on Cell Imaging 24-well Plates (Eppendorf, Hamburg, Germany) in 500 µL of the experimental medium at 37 °C and 5% CO_2_. After 24 h, cells were transfected as described above, and the day after time-lapse imaging was performed using Operetta CLS™ (PerkinElmer, Inc, Waltham, MA, USA) equipped with a 63× immersion objective. The instrument was set up to capture images of each chosen field every 2 s, in brightfield and green, or red fluorescent channels, for 10 min at 37 °C and 5% CO_2_.

### 2.8. Western Blot Analysis

Western blot analyses were performed as previously described [40]. Briefly, signals were detected using peroxidase-conjugated secondary antibodies; and immunoblots were developed using ECL Westar Nova 2.0 detection system (Cyanagen, Bologna, Italy). Original western blots are included in Appendix A.

### 2.9. Co-Immunoprecipitation Assay

HEK293 and 5637 cells were seeded at the density of 3 × 10^6^ cells/dish in 100 mm dishes (Euroclone, Pero, Italy) in DMEM or RPMI-1640 with 10% FBS. The next day they were serum-starved (0.5% FBS) for 18 h and later stimulated with 10% FBS for 15 min (S) or left unstimulated (NS). Co-immunoprecipitation assays were performed as previously described [40].

### 2.10. Stress Media Composition and Assay

To determine the average number of cells forming TNTs under several growth conditions, 5637 and HEK293 cells were cultured using: (I) High Serum Medium (HS) composed of DMEM medium supplemented with 10% FBS at normal pH 7.4 with 25 mM glucose (usually referred to as standard culture medium); (II) Low Serum Medium (LS) composed of DMEM with 2.5% FBS at normal pH 7.4 with 25 mM glucose; (III) Low Serum Acidified Medium (LSA) composed of DMEM medium supplemented with 2.5% FBS at low pH (6.6) with 50 mM glucose [19]; (IV) High Serum Medium with hydrogen peroxide (HS+H_2_O_2_) composed of DMEM with 10% FBS at normal pH 7.4 with 25 mM glucose and 100 µM H_2_O_2_; (V) Low Serum Medium with hydrogen peroxide (LS+H_2_O_2_) composed of DMEM with 2.5% FBS at normal pH 7.4 with 25 mM glucose and 100 µM H_2_O_2_ [49]. Cells grown in LSA, HS+H_2_O_2_ and LS+H_2_O_2_ media were first tested at different time points (3, 24, 48 and 72 h; untreated controls are referred to as NT) to analyze cell morphology, membrane protrusion formation and loss of adherence; thus, ensuring that the generated stress did not result in cell death. LSA and HS+H_2_O_2_ were also used to treat 5637 and HEK293 cells to evaluate RalGPS2 expression and p53 phosphorylation levels, respectively. Briefly, HEK293 and 5637 cells were plated in 100 mm dishes (Euroclone, Pero, Italy) in DMEM or RPMI-1640 medium supplemented with 10% heat-inactivated FBS. The day after, the medium was replaced with LSA or HS+H_2_O_2_, and, after 24 h, the cells were lysed, and western blot analysis was performed as described above. Each experiment was performed in triplicate. Data were analyzed using GraphPadv6.0 software (San Diego, CA, USA) employing either ANOVA followed by Tukey’s test for group comparison or Student’s *t*-test. *p* < 0.05 was considered statistically significant.

### 2.11. RNA Interference

For siRNA experiments, 5637 and HEK293 cells were plated at a density of 1 × 10^5^ cells/mL in 6-well plates (Euroclone, Pero, Italy) in RPMI-1640 or DMEM medium with 10% FBS. The next day, cells were co-transfected with pEGFP-C1 vector and 25 nmol RalGPS2-specific Stealth siRNA, as previously described [40]. Then the medium was replaced with LSA or HS+H_2_O_2_ and, after 24 h, the cells were tested. Data were analyzed using GraphPadv6.0 software (San Diego, CA, USA) employing Student’s *t*-test. *p* < 0.05 was considered statistically significant.

### 2.12. TNTs Analysis under Pharmacologically Perturbed Conditions

5637 and HEK293 cells were plated at a density of 1 × 10^5^ cells/mL on porcine gelatin pre-treated coverslips and transfected for RNA interference as described above or left un-transfected. The next day, the medium was replaced with LSA or HS+H_2_O_2_, and after 24 h, the cells were treated with or without 100 nM Wortmannin (Sigma-Aldrich, St. Louis, MO, USA) or 10 µM A6730 (Sigma-Aldrich, St. Louis, MO, USA) for 1 h. After treatment, cells were stained with DiI (Sigma-Aldrich, St. Louis, MO, USA) to label cell membranes, according to the manufacturer’s instructions. Cells were then fixed for 10 min with 3.7% paraformaldehyde in phosphate-buffered saline (PBS). Fluorescence images were obtained with Leica TCSSP2 confocal microscope (Leica, Wetzlar, Germany) equipped with a 63×/1.4 NA Plan-Apochromat oil immersion objective. To quantitative determination, the percentage of cells forming nanotubes was assessed. About 200 cells, derived from different view fields of a plate, were analyzed for each experiment. The analysis was performed at the magnification described above. Experiments were performed in triplicate. Data were analyzed using GraphPadv6.0 software (San Diego, CA, USA) employing ANOVA followed by Tukey’s test for group comparison. *p* < 0.05 was considered statistically significant.

## 3. Results

### 3.1. Tunneling Nanotubes Formation in Bladder Cancer Cell Lines

Intercellular transfer of proteins [50,51], microRNA [17] and organelles [12,52] via tunneling nanotubes (TNTs) connecting long-distance cells promotes cell proliferation and invasion [16,53]. The type of TNTs formed by cells is affected by cell density, namely the distance and proximity between cells [54]. 

To characterize the bladder cancer microenvironment affected by TNTs, we analyzed a panel of six bladder cancer cell lines at different stages and grades, including RT4, RT112, 5637, HT1376, UMUC3 and J82 cell lines [48], which showed varying degrees of genetic complexity and different invasive properties. We assessed the number of cells forming tunneling nanotubes, the expression of proteins involved in tunneling nanotubes formation and the transfer of mitochondria via TNTs.

As shown in Figure 1, low-stage cell lines RT4 and RT112 do not exhibit TNTs, while medium and high-stage muscle invasive cell lines 5637, HT1376, UMUC-3 and J82, generate TNTs forming a complex network of cell-to-cell connections.

The percentage of cells forming TNTs in the different cell lines was assessed (Figure 2). Approximately 30% of cells in 5637 cell lines are able to make TNTs, while only approximately 20% of cells in UMUC-3 and J82 cell lines produce them. HT1376 cell lines display the lowest percentage of cells forming TNTs, only 10%.

### 3.2. Mitochondrial Transfer in Bladder Cancer Cell Lines through TNTs

Tunneling Nanotubes (TNTs) are open-ended channels. In cancer, they drive the malignant phenotype [55], for example, allowing mitochondrial exchange [52,56].

To assess if TNTs observed in mid and high-stage bladder cancer cells show functional mitochondrial trafficking, we labeled cells with MitoTracker Green, a green-fluorescent mitochondrial dye that localizes to mitochondria regardless of mitochondrial membrane potential. We then performed live imaging analysis using an automated screening microscope, Operetta CLS™.

As shown in Figure 3 and Appendix A, all cell lines considered were able to exchange mitochondria via TNTs. For most cell lines, the percentage of cells that transferred mitochondria was under 10%, with the exception of UMUC-3 cells [57], which have the most invasive phenotype, reaching 30% (Figure 4). Furthermore, to understand if mitochondria transferred via tunneling nanotubes were functional, cells were labeled with MitoTracker Green in combination with Tetramethylrhodamine, Ethyl Ester, Perchlorate (TMRE). TMRE is a red-orange fluorescent dye that is quickly sequestered by active mitochondria. No correlation between mitochondrial trafficking and functionality during transport was reported (Appendix A). Our data show that bladder cancer cells can exchange both functional and non-functional mitochondria.

Since TNTs were not detected in RT4 and RT112 cell lines, we did not analyze them in this experiment. Our data demonstrate that nanotubes generated by bladder cancer cells are functional.

### 3.3. Expression of Proteins Involved in Tunneling Nanotubes Formation in Bladder Cancer Cells

RalA GTPase is involved, together with the exocyst complex, in tunneling nanotubes formation [44]. It has been previously demonstrated that RalGPS GEFs have a prominent role in RalA activation in 5637 bladder cancer cell lines. Indeed, on the one hand, silencing RalGPS2 inhibits endogenous nanotubes formation; on the other hand, its over-expression, or the downregulation of RalGDS family proteins caused by the expression of the dominant-negative mutant H-RasV12S35, which specifically activate Raf and not RalGDS family of GEFs [58], increases the length and the number of TNTs [40].

To study the expression of RalA GTPase and RalGPS2 in the six bladder cancer cell lines previously considered, western blot analyzes were performed. GAPDH was used as the loading control to normalize and quantify protein amount. RalGPS2 is expressed in all cell lines, though very low levels of expression were detected in UMUC-3 and J82 cell lines (Figure 5A,B). Only in the 5637 cell line two isoforms of RalGPS2 were detected. RalA was expressed at a high level only in 5637 and HT1376 cell lines (Figure 5A,C).

The 5637 cell line showed the highest percentage of nanotubes-forming cells and the highest expression of RalA and RalGPS2. Consequently, they were selected as a model to perform further analyzes aimed at studying the TNTs microenvironment in bladder cancer cells.

### 3.4. RalA and LST1 Are Transferred via TNTs in 5637 Cells

The transfer of proteins through TNTs is a well-documented mechanism [50,51]. Since 5637 cells showed functional TNTs able to transfer mitochondria, their ability to transfer RalA GTPase and RalGPS2 was assessed. 5637 cells were transfected either with RalA-mCherry or RalGPS2-GFP constructs, and the day after transfection, live imaging analysis was performed using Operetta CLS™. RalA GTPase pass through TNTs, while RalGPS2 was not detectable in tunneling nanotubes (Figure 6A,C and Appendix A).

Another important molecule involved in TNTs formation is leukocyte-specific transcript 1 (LST1), a well-established binding partner of RalA and RalGPS2 [36]. To investigate if LST1 is transported via tunneling nanotubes in our model, 5637 cells were transfected with LST1-mCherry construct and live image analysis was performed, as described above. As shown in Figure 6B, LST1 was effectively transferred via TNTs in 5637 cells (Appendix A).

### 3.5. Stress Conditions Promote TNTs formation in HEK293 and 5637 Cells

Wang et al., in 2011, demonstrated that cells in induced stress conditions produce more nanotubes compared to cells cultured in physiological conditions [49]. Specifically, Wang et al. demonstrated that treatment with H_2_O_2_ and serum depletion induces TNTs formation in astrocytes and neurons. TNTs formation is thus induced by different cellular stressors, such as nutrient deprivation, low pH, hypoxia and oxidative stress [59].

To assess which stress condition, among low-serum, acidification or oxidative insult, could promote TNTs formation in 5637 and HEK293 cells, we tested different culture media (Materials and Methods) mimicking the diverse types of stress.

HEK293 cells were choosen as a cellular model. to validate our findings in other cells of the urinary system. In this cell line, RalGPS2 is expressed and able to activate RalA GTPase, as our group previously demonstrated [45]. In all culture conditions tested, we assessed cell responses, such as morphology changes and membrane protrusions formation.

HEK293 and 5637 cells are normally grown in DMEM and RPMI-basedmedium, respectively, containing 25 mM glucose, 2 mM L-Glutamine and 10% FBS, and the pH of the culture medium is buffered to 7.4.

To induce stress conditions, we decided to use two media previously described in the literature [19,49]. The first one, a low-pH medium, referred to as LSA, is an Acidified hyperglycemic Low-Serum medium (pH 6.6, 50 mM glucose, 2.5% foetal bovine serum) that boosts the invasive potential of cancer cells (Appendix A); the second, here referred to as H_2_O_2_ (tested in two different serum conditions: High Serum (HS containing 10% foetal bovine serum) or Low Serum (LS containing 2.5% foetal bovine serum)), is an oxidative stress medium (DMEM with 10% or 2.5% FBS at normal pH 7.4 with 25 mM glucose and 100 µM H_2_O_2_) that induces cellular insult and leads to p53 activation (Appendix A). Since maintaining cells in low serum is a well-known TNTs inducer [19,60], we decided to test this experimental setting in combination with H_2_O_2_ to enhance nanotubes generation.

As shown in Appendix A, the formation of TNTs increased in 5637 cells 24 h after LSA medium addition, while HEK293 cells died after 180 min.

The H_2_O_2_-containing medium caused no effect in 5637 cells. Indeed, even after 72 h of exposure to H_2_O_2_ medium, the cells continued to grow, and no morphological changes were visible, both in the HS and LS media (Appendix A).

HEK293 cells responded to oxidative stress producing a higher number of TNTs compared to control, already at 24 h in HS medium, while in the LS medium, the cells showed signs of suffering and no TNTs was detected. At 72 h in the HS medium, TNTs increased, but cells showed morphological differences: they became rounded and partially detached from the substratum. Moreover, in the LS medium, cells died, indicating that the stress level was too high.

Therefore, our results demonstrate that TNTs production in 5637 and HEK293 cells is increased by acidification and oxidative stress, respectively.

To confirm the data obtained by the inverted microscope, we analyzed TNTs formation by confocal microscopy, comparing cells cultured in HS and LS media (10% or 2.5% FBS) to cells in an appropriate stress medium.

In the HS medium, 5637 cells showed a higher percentage (~30%) of cells forming TNTs than the HEK293 cell line (~10%) (Figure 7B,D). Furthermore, there was no difference in the percentage of HEK293 cells forming TNTs in HS or LS media. Instead, 5637 cells cultured in LS reached 50% of cells displaying TNTs. Thus, low-serum conditions represent a stressful state for 5637 cells, while HEK293 cells are not affected.

As shown in Figure 7, LSA and HS+H_2_O_2_ media increased the percentage of cells forming TNTs up to 80%, confirming the data obtained by inverted microscope.

### 3.6. RalGPS2 Is Crucial to TNTs Formation Also in Stress Conditions in 5637 and HEK293 Cells

We have previously demonstrated that RalGPS2 is involved in the formation of TNTs and that the overexpression of RalGPS2 markedly increases the number and length of nanotubes [40,61].

Furthermore, we have demonstrated that the knock-down of RalGPS2 using siRNA leads to a strong reduction of TNTs formation in 5637 cells [40]. Therefore, we decided to test the effect of RalGPS2 depletion combined with TNTs induction promoted by stress media (LSA and HS+H_2_O_2_).

After transfection with the specific siRNA for RalGPS2, cells were cultured for 24 h in LSA and HS+H_2_O_2_ media.

RalGPS2 silencing caused a strong reduction in the percentage of cells producing TNTs in both cell lines (Figure 8). These data suggest that RalGPS2 has a crucial role in TNTs formation.

Subsequently, to further evaluate if the increase in TNTs formation observed under stress conditions was related to an enhancement in RalGPS2 expression, we performed western blot assays after 24 h from stress medium administration. Cells in HS medium were used as control. The experiment was carried out using a specific stress medium for each cell line. As it is shown in Figure 9A,B that RalGPS2 expression increased in both cell lines at 24 h after exposure to stress conditions.

The p53 tumor suppressor has a vital role in regulating cancer cell death, cell cycle arrest, apoptosis, senescence and DNA repair [62]. Moreover, p53 responds to various stress signals promoting the transcription of a series of genes and is critical for TNTs formation in astrocytes under stress conditions [49].

To analyze p53 activation (through assessment of its phosphorylation status), HEK293 and 5637 cells were cultured in specific stress media, and equal amounts of total protein extracts were loaded onto 8% denaturing polyacrylamide gel. Active p53 was assessed by western blot assays, using a specific antibody that detects phosphorylation occurring on Threonine 155 residue of p53. For both cell lines, the active-p53 expression did not change (Figure 9C,D). In conclusion, data obtained using confocal microscopy and western blot assays show that an increase in TNTs formation is related to an enhancement in RalGPS2 expression, but not to a change in the levels of phospho-p53 (p53 P).

### 3.7. PI3K and Akt Inhibition Reduces TNTs Formation in HEK293 and 5637 Cells

Upregulation of the Akt/PI3K/mTOR signaling pathway by stress conditions is a well-documented mechanism involved in nanotubes formation [19]. In fact, under stressful conditions, p53 up-regulates EGFR and TNF⍺-induced protein 2 (TNFαIP2, also known as M-Sec), which in turn activates Akt/PI3K/mTOR and RalA-exocyst pathways, respectively. Both pathways are necessary for TNTs development [49]. D’Aloia et al., in 2018, showed that RalGPS2 affects Akt activation in 5637 cells [40]. Therefore, we decided to further investigate, both in HEK293 and 5637 cells, whether RalGPS2 interacts with Akt and the protein kinase responsible for Akt activation, pyruvate dehydrogenase kinase 1 (PDK1) [40].

Using co-immunoprecipitation assays, we demonstrated that both active/phosphorylated (Akt P) and total Akt and PDK1 co-immunoprecipitated with RalGPS2. Experiments were performed using serum-starved cells (0.5% FBS), following either 15 min of serum stimulation (S = stimulated with 10% FBS), or as control, leaving them unstimulated (NS = not stimulated) (Figure 10). With our experiments, we could demonstrate that RalGPS2 is able to form a complex with Akt and PDK1, as tested in HEK293 and 5637 cell lines. Within this complex, p53 P and TNFαIP2 are also present.

To elucidate the mechanisms and upstream signals that mediate RalGPS2 activation, we studied which consequences might have the inhibition of proteins involved in the complex activation. Since the PH domain of RalGPS2 binds to phosphatidylinositol 4,5 bisphosphate (PIP2) [45], a PI3K product, we decided to test the effect of wortmannin, a PI3K inhibitor. HEK293 and 5637 cells were grown in the specific stress medium for 24 h; then, wortmannin was added for 1 h. Cells were stained with the lipophilic membrane stain DiI and assessed for TNTs presence.

PI3K inhibition caused a decrease in the number of cells with TNTs, resulting in a percentage of cells having TNTs similar to the ones observed in cells grown in HS medium (Figure 11). Furthermore, PI3K inhibition in HEK293 cells, similarly to RalGPS2 silencing, diminished the proportion of cells forming TNTs, while in 5637 cells, RalGPS2 depletion had a more pronounced effect than PI3K blockage.

Subsequently, we decided to test the effect of A6730, an Akt inhibitor, on tunneling nanotubes formation.

The number of cells forming TNTs was reduced in both cell lines. In the HEK293 cell line, a lower percentage of TNTs-displaying cells was observed, though the number was higher compared to cells in the control condition and under treatment with wortmannin. Taken together, our results suggest that although Akt is important in TNTs generation, it is not essential in this process. Conversely, the presence of an active RalGPS2 pathway is crucial.

In the 5637 cell line, the percentage of cells forming nanotubes is comparable to the one resulting upon PI3K inhibition, indicating the Akt pathway plays a fundamental role in the process.

### 3.8. RalGPS2 Forms a Complex with LST1, RalA and Sec5 in HEK293 Cells

In the 5637 cell line, RalGPS2 forms a complex with RalA, Sec5 and LST1 that is required for TNTs formation, as we previously demonstrated [36,40]. Here, we evaluated if this complex is also present in HEK293 cells. Co-immunoprecipitation assays (RalGPS2, LST1 and Sec5) were performed on serum-starved (0.5%FBS) HEK293 cells under serum-stimulated (S) or unstimulated (NS) conditions. We found that, also in HEK293 cells, RalA co-immunoprecipitated with RalGPS2, Sec5 and LST1 and these three proteins interacted with each other (Figure 12). Furthermore, no difference was observed between the stimulated or unstimulated conditions, and in the immunoprecipitation complex, we found both the trimeric and the monomeric form of LST1, as it has been shown for 5637 cells in D’Aloia et al., 2018 [40]. Thus, the complex first identified in 5637 bladder cancer cells is also present in other cell types, such as HEK293.

## 4. Discussion

The tumor microenvironment plays a fundamental role in cancer progression and resistance [63]. Tumors are not merely a mass of cancer cells; they are also a heterogeneous environment where cancer cells recruit stromal, immune system and endothelial cells by secreting growth factors and cytokines. In turn, the recruited cells, such as cancer-associated fibroblasts (CAFs) [64,65], secrete signals and metabolites that support the growth of cancer cells. The communication between these cells in the tumor mass is not due exclusively to paracrine signaling, but it is also based on a network of membranous protrusions, the tunneling nanotubes. Through TNTs, cells can directly exchange small molecules and organelles [66], providing them with new traits, such as chemoresistance, migratory phenotype and metabolic plasticity [67]. Functional TNTs were observed in several types of cancer and were also imaged in intact tumors, proving their potential in vivo relevance [19,68].

Originally reported in 2004 by Rustom et al. in rat Pheochromocytoma (PC12), Human Embryonic Kidney (HEK293) and Normal Rat Kidney (NRK) cells [9], TNTs are highways for intracellular organelles and proteins. They act as an inter-cellular transport system. They have subsequently been identified in several eukaryotic cells, including urothelial [33,69], neuronal, immune [15] and primary cells [70].

Similar structures have been found in bacteria. Such structures allow the production of antibiotics, secretion of virulence factors and bioluminescence [71].

RalA overexpression is associated with several different types of human cancers, in particular bladder cancer [72], which represents a major epidemiological problem and whose incidence continues to increase each year.

In our study, we focused our attention on the bladder cancer microenvironment, using a panel of six cell lines representative of bladder cancers at different stages and grades. We demonstrated that TNTs formation correlates with the stage: active TNTs, able to transfer mitochondria, are formed only between cells at mid and high stages. Usually, the transfer of mitochondria through tunneling nanotubes occurs preferentially from endothelial cells to cancer cells, for example, by conferring them chemoresistance [73]. However, several studies report that the exchange also takes place between cells of the same type [18,49].

Mitochondria are the powerhouses of the cell, and their acquisition by a donor cell confers a metabolic potential capable of guaranteeing significant survival benefits under stressful conditions. In fact, under nutritional shortness, mitochondrial respiration enables a high yield of ATP production per unit of moles of consumed glucose. This, in turn, ensures the cell can get energy from the oxidation of other carbon sources, such as amino acids, fatty acids and lactate, which can also support the anaplerosis in hyper-glycolytic cells where glucose follows a fermentative route (aerobic glycolysis, or Warburg effect).

Therefore, the transfer of mitochondria can supply energy and metabolic support to tumor cells, possibly mitochondria-defective cells, allowing aggressive phenotypes typical of different kinds of cancer [52,56]. Mitochondria uptake could also support cell resistance to stress by promoting aerobic respiration, decreasing ROS production and increasing survival [74,75,76,77].

To further characterize TNTs in the six bladder cancer cell lines under investigation, we analyzed the expression of proteins involved in TNTs generation: namely, RalA GTPase and RalGPS2 [40,44]. RalA contributes to the control of actin cytoskeletal remodeling and vesicular transport via interaction with the exocyst complex [78]. Previous work suggests that RalA may coordinate with M-Sec and exocyst to initiate TNTs formation and elongation [44]. RalGPS2 is a Ras-independent GEF for RalA, which contains a PH domain and a SH3-binding region, and it is involved in cytokinesis, the control of cell cycle progression, differentiation, organization and rearrangement of cytoskeleton and TNTs development in PC12 and 5637 cells [40,45,61]. Moreover, the *RALGPS2* gene is regulated by HIF-2α and directly affects endothelial sprouting during prolonged hypoxic culturing [79].

The expression of RalGPS2 and RalA is very low in UMUC-3 and J82 high-stage and grade cell lines, while it is very high in 5637 and HT1376 cells. Among the bladder cancer cell lines studied in our work, only UMUC-3 expresses an oncogenic mutant of KRas, namely KRasG12C [48,80,81,82]. Thus, although UMUC3 cells display the highest percentage of functional TNTs, the presence of KRas mutation does not make them suitable as model cell lines to study the TNTs microenvironment in bladder cancer cells. Indeed, hyperactivated Ras directly affects RalA activity [83,84].

Smith and colleagues, in 2007, demonstrated, in bladder cancer, that the overexpression of RalA is associated with tumors of high-stage and grade. However, RalA protein has a lower expression in UMUC3 and J82 cells compared to RT4 and 5637 cells. These apparently conflicting results can be explained by the fact that despite lower levels of total RalA, high-stage and grade cells contain a higher fraction of the protein in the active state, as demonstrated in J82 cells [85]. Analysis of human bladder cancer cell lines also revealed RalA activation, which cannot be caused by protein mutations affecting its intrinsic GTP-binding activity [86].

The presence of multiple isoforms of RalGPS2 protein may be due to alternative splicing or post-translational modification variants [45,87]. This aspect had already been reported in different mouse tissues [45]. In conclusion, here we show that there is no correlation between the expression levels of RalA and RalGPS2 and the formation of tunneling nanotubes.

TNTs can transfer proteins. For example, TNTs connecting B and T cells transfer H-Ras protein from B to T cells [51]; in addition, TNTs facilitate the direct intercellular transfer of oncogenic K-Ras between colorectal cancer cells [50]. TNTs observed in 5637 cells are functionally active and involved in mitochondrial trafficking; here, we demonstrated that TNTs in bladder cancer cells are also able to transport molecules important for TNTs development, such as LST1 and RalA, but not RalGPS2. Our work provides, for the first time, evidence that TNTs transfer RalA GTPase from cell to cell.

Taking into account that RalA is necessary for the anchorage-independent growth of cancer cells and its overexpression is associated with several different types of human tumors, our data are of particular interest [85].

Different studies have demonstrated that stressful states (e.g., inflammation, high glucose and low pH growth medium, low serum, ultraviolet radiation, temperature, H_2_O_2_) can boost TNTs generation between cells [18,19,88,89]. Furthermore, previous works have shown that under stress conditions, cells release proteins or metabolites into the culture medium. Such released factors can act as ‘call-for-help’ signals for other cells, leading to nanotubes formation [90]. In the present work, we show that TNTs are formed spontaneously during in vitro growth: however, they are most abundant in acidified low-serum medium (LSA) or in oxidative stress medium (HS+H_2_O_2_) in 5637 and HEK293 cells, respectively. Notably, LSA mimics the acidified tumor microenvironment deriving from the fermentative activity of hyperglycolytic proliferating cancer cells. This condition is known to support the invasive potential of cancer cells, most likely contributing to TNTs formation [19,91]. Conversely, hydrogen peroxide (H_2_O_2_), one of the most abundant oxygen reactive species (ROS), has been demonstrated to induce cell injury through lipid oxidation and DNA damage triggering cellular apoptotic pathways [92,93,94,95,96,97]. In addition to these effects, H_2_O_2_ promotes actin polymerization, co-localization of myosin Va with F-actin and tunneling nanotubes formation among astrocytes [98]. In this scenario, with our work, we demonstrate that the enhancement of TNTs generation caused by stressful media is the consequence of RalGPS2 up-regulation. Indeed, the knock-down of RalGPS2 inhibits TNTs development. Taken together, our findings pinpoint RalGPS2 as a key regulator of nanotubes formation.

Another important regulator of TNTs generation is p53. Wang et al., in 2011, have demonstrated that hydrogen peroxide (H_2_O_2_) induces p53 activation (phospho-p53), which in turn boosts TNTs development between primary rat hippocampal astrocytes and neurons [49]. Moreover, other studies demonstrated that p53, as well as CDC42, myosin X (Myo-D), M-Sec (also called B94 or tumor necrosis factor-induced protein 2; TNFαIP2), RalA-GTP, LST1 and exocyst complex are important regulators of nanotubes formation [49,60,99,100]. In the present study, we showed that even though phospho-p53 (p53 P) interacts with RalGPS2, stressful conditions modulate only RalGPS2 expression, while p53 activation is not affected. However, though phospho-p53 levels do not seem to change until the 24 h timepoint, it interacts with RalGPS2, as well as with TNFαIP2. These findings suggest a possible role of phospho-p53 and TNFαIP2 in the RalGPS2 activation complex involved in TNTs generation. In fact, Wang et al. demonstrated that under stressful states, active p53 up-regulate both EGFR and TNFαIP2 [49].

Among the pathways involved in nanotubes development, the Akt/PI3K/mTOR signaling pathway plays a crucial role. In fact, in human malignant pleural mesothelioma cell lines, hyperactivation of this signal cascade by low-serum growth medium promotes TNTs formation [19]. The Akt signaling pathway is involved in the regulation of F-actin polymerization, production of cellular protrusions, cell polarization and adhesion [101,102,103], which are necessary to form mature and stabilized TNTs [104]. Furthermore, our group has previously demonstrated that RalGPS2 regulates Akt activation [40]. Notably, the silencing of RalGPS2 partially inhibits Akt phosphorylation, while its overexpression increases it [40]. In the current work, we show that the ability of RalGPS2 to modulate Akt activation is promoted by the interaction between RalGPS2, Akt (total and active Akt) and PDK1 kinase. PDK1 is the protein kinase responsible for Akt phosphorylation and activation in the plasma membranes [105]. These findings, together with those obtained by D’Aloia et al., in 2018 [40], suggest that RalGPS2 may contribute to Akt activation, acting as a scaffold protein and enabling PDK1 to come within close proximity with Akt. Similar results have already been observed for the Ras-dependent Ral GEFs family, RalGDS. Indeed, RalGDS promotes Akt phosphorylation via PDK1 by bringing these two kinases together [106].

RalGPS2, as described by Ceriani et al., is a RalA GEF containing a Cdc25-like GEF domain and a pleckstrin homology (PH) domain in the C terminus. The latter binds phosphatidylinositol-4,5-bisphosphate (PIP_2_) but, at a lower extent, also binds phosphatidylinositol 3-4-5-triphosphate (PIP_3_), both are PI3K products [45]. Moreover, phosphatidylinositol 3-kinase (PI3K) is the protein kinase that, together with PDK1, contributes to Akt activation. The most well-established model of Akt phosphorylation involves ligand-induced activation of PI3K, which in turn leads to the production of the phospholipids PIP_2_ and PIP_3_ [107]. These phospholipids interact with the PH domains of Akt and PDK1 in the plasma membrane, where PDK1 phosphorylates Akt on threonine 308 [105]. In this work, we demonstrate that the use of wortmannin, a potent, selective and irreversible PI3K inhibitor, disrupts TNTs development both in 5637 and HEK293 cell lines, although these cells had been treated with TNTs-inducer media (stress media). Our findings prove the prominent role of PI3K in TNTs formation. Our hypothesis is that the inhibition of PI3K by wortmannin impairs RalGPS2 localization in plasma membranes. Indeed, despite the fact that stress conditions can increase RalGPS2 expression, the blockage of PI3K activity inhibits the production of PIP_2_ and PIP_3,_ and consequently, the anchorage of RalGPS2 at the plasma membrane. Moreover, we demonstrated that Akt activation is essential for TNTs formation in 5637 cells, as the use of a selective Akt inhibitor (A6730) impairs TNTs generation. Conversely, the Akt pathway does not play an important role in nanotubes development in HEK293 cells. This is in agreement with the notion that different mechanisms are present, which promote TNTs formation in several types of cells.

D’Aloia et al. demonstrated that RalGPS2 interacts with LST1 and RalA, leading to the formation of a complex that promotes TNTs generation in 5637 cells [40]. In the current work, we have shown that this complex is also formed in HEK293 cell lines, and it is likely to be responsible for TNTs development in these cells.

Taken together, our results extend the molecular machinery shown by D’Aloia et al. [40]. In this study, stress conditions boost RalGPS2 expression and induce PI3K activation, which in turn leads to the production of PIP_2_ and PIP_3_. These phospholipids are substrates of the PH domain of Akt, PDK1 and RalGPS2 in the plasma membrane, where RalGPS2 may contribute to Akt activation, acting as a scaffold protein for PDK1 and Akt. In this context, PDK1 phosphorylates Akt and in turn activates the Akt signal cascade, which controls TNTs formation through mTOR pathways, or cell proliferation, through p21 and p27 inhibition. Moreover, in this scenario, RalGPS2, recruited to the plasma membrane by PI3K products, also interacts with phospho-p53, TNFαIP2, LST1 and RalA, leading to the assembly of a specific multimolecular complex. Under a stressful state, this complex allows RalA-Sec5 interaction and activation (assembly of exocyst complex), which prompts TNTs development (graphical abstract).

## 5. Conclusions

The exchange of material through TNT connections gives cancer cells, in the context of the tumoral microenvironment, specific traits (e.g., enhanced metabolic plasticity, migratory phenotypes, angiogenic ability and therapy resistance). In this paper, we demonstrate that TNTs formed by bladder cancer cells of mid and high-stages with invasive properties [108] are functionally active and transport organelles, such as mitochondria, and proteins involved in bladder cancer malignancy and in nanotubes formation, such as RalA GTPase and LST1. Furthermore, we characterized the molecular machinery underlying TNTs formation in bladder cancer and kidney cells and unveiled the crucial role played by RalGPS2, which promotes tunneling nanotubes formation through interaction with Akt and PDK1.

We also demonstrate that TNTs formation is induced upon a specific stress condition. The diversity in specific stress stimuli considered may reflectthe different roles played by some molecules involved in TNTs development. For instance, in HEK293 cells, TNTs are formed in response to oxidative insults: RalGPS2 is essential in the process, while Akt is dispensable. Conversely, in 5637 cells, TNTs are formed in response to medium acidification and serum limitation: in this model, RalGPS2, as well as Akt, are crucial for TNTs formation. Given the crucial role of TNTs in tumor progression, TNTs-interfering drugs represent a new frontier to reduce cellular invasiveness and pave a new way towards clinical application. Clearly, this gives an idea of how the development of precision medicine is essential in the treatment of certain types of diseases, such as cancer, where patients must be stratified based on markers to be subjected to more effective and less toxic targeted therapies. In this study, RalGPS2 emerges as a potential common pharmacological target for the development of TNTs-interfering drugs that can reduce tumor progression, at least in bladder cancer, by blocking supportive intercellular mechanisms within the tumor microenvironment.

## Figures and Tables

**Figure 1 cancers-13-06330-f001:**
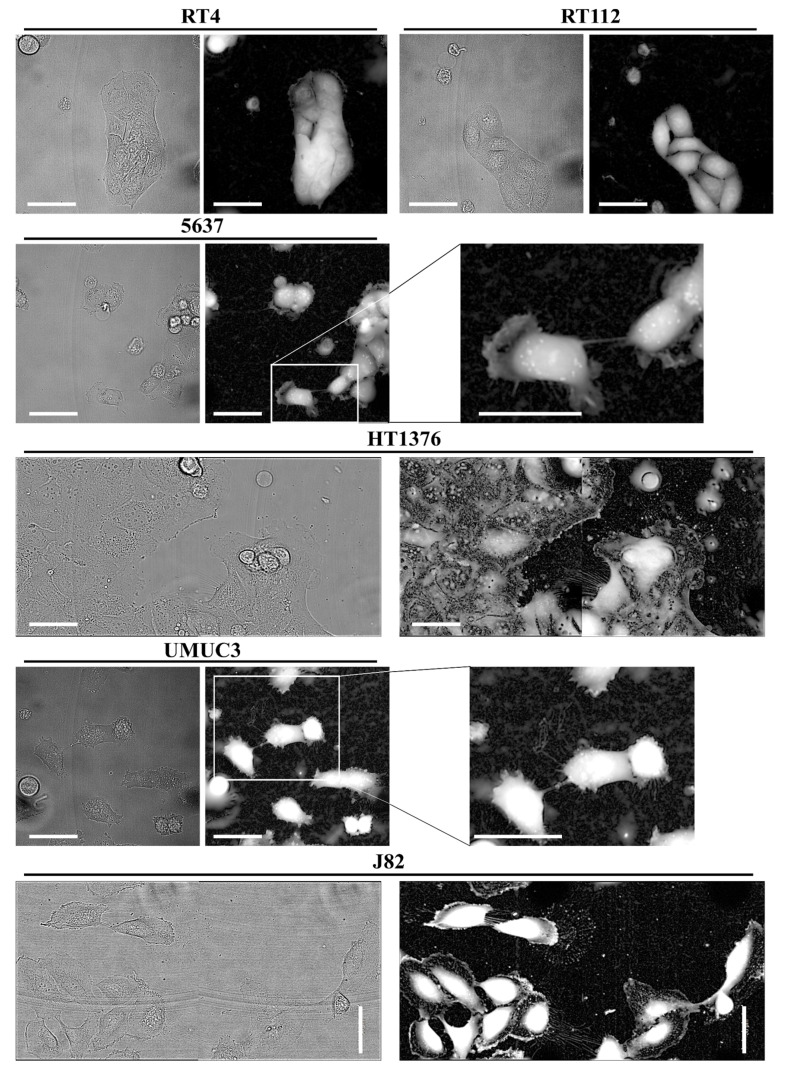
Tunneling nanotubes formation in bladder cancer cell lines. RT4, RT112, 5637, HT1376, UMUC-3 and J82 cells were plated at a density of 1 × 10^4^ cells/well on CellCarrier Ultra 96-well microplates. After 24 h, live cell images were acquired using Operetta CLS™ equipped with 63× immersion objective in brightfield (left panels) and Digital Phase Contrast (DPC) (right panels). Scale bar: 50 µm. Magnifications show TNTs.

**Figure 2 cancers-13-06330-f002:**
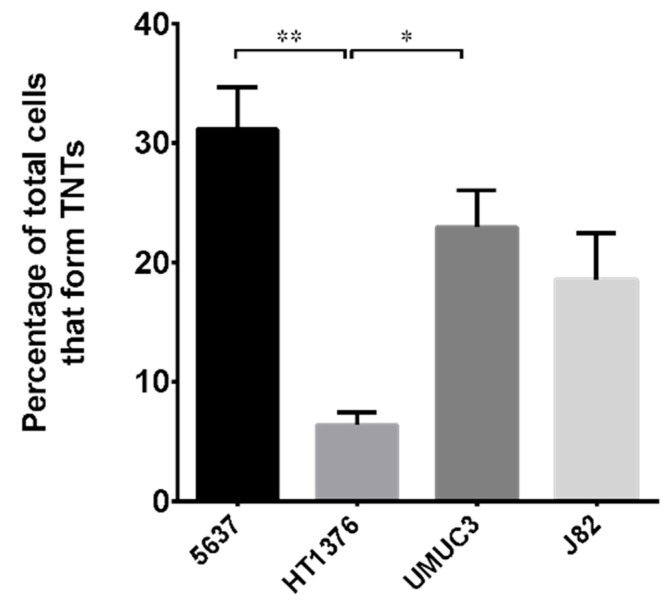
Percentage of cells forming TNTs in different bladder cancer cell lines. 5637, HT1376, UMUC-3 and J82 cell lines were plated as described in Figure 1, imaged by Operetta CLS™ and scored for the presence of nanotubes. At least 200 cells were analyzed per group in three independent experiments. Data are expressed as mean ± S.E.M. from three independent experiments. Differences among groups were analyzed using a one-way analysis of variance (ANOVA) followed by Tukey’s post hoc test. * *p* < 0.05, ** *p* < 0.01.

**Figure 3 cancers-13-06330-f003:**
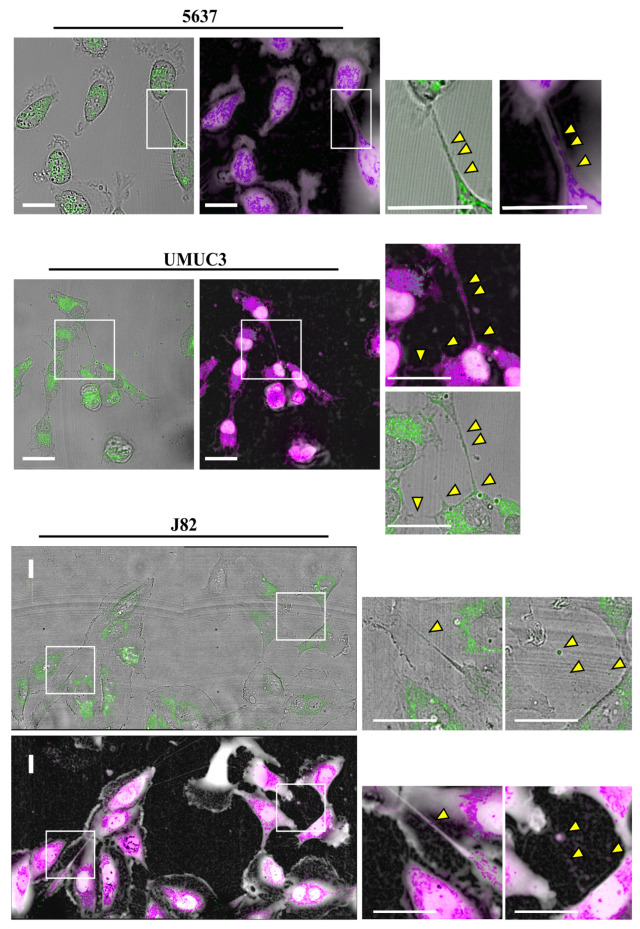
Mitochondrial transfer via TNTs in bladder cancer cell lines. 5637, UMUC-3 and J82 cells were plated at a density of 1 × 10^4^ cells/well on Cell Imaging 24-well Plates. After 24 h, cells were stained with MitoTracker™ Green and live cell images were acquired using Operetta CLS™, equipped with 63× immersion objective in brightfield, DPC and fluorescence, to detect MitoTracker. Scale bar: 20 µm. Yellow triangles indicate mitochondria. Magnified views of mitochondria inside nanotubes are shown.

**Figure 4 cancers-13-06330-f004:**
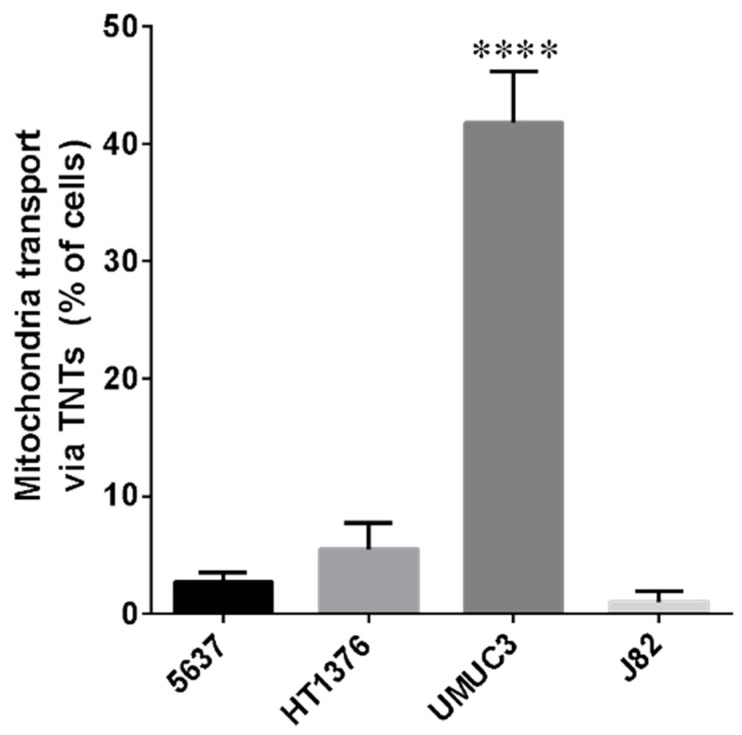
Percentage of bladder cancer cells able to exchange mitochondria via TNTs. 5637, HT1376, UMUC-3 and J82 cells were plated and stained as described in Figure 3, imaged by Operetta CLS™ and scored for the presence of mitochondria inside TNTs. At least 200 cells were analyzed per group in three independent experiments. Data are expressed as mean ± S.E.M. from three independent experiments. Differences among groups were tested for significance by the one-way analysis of variance (ANOVA) followed by Tukey’s post hoc test. **** *p* < 0.0001 indicated significance compared to all the other groups.

**Figure 5 cancers-13-06330-f005:**
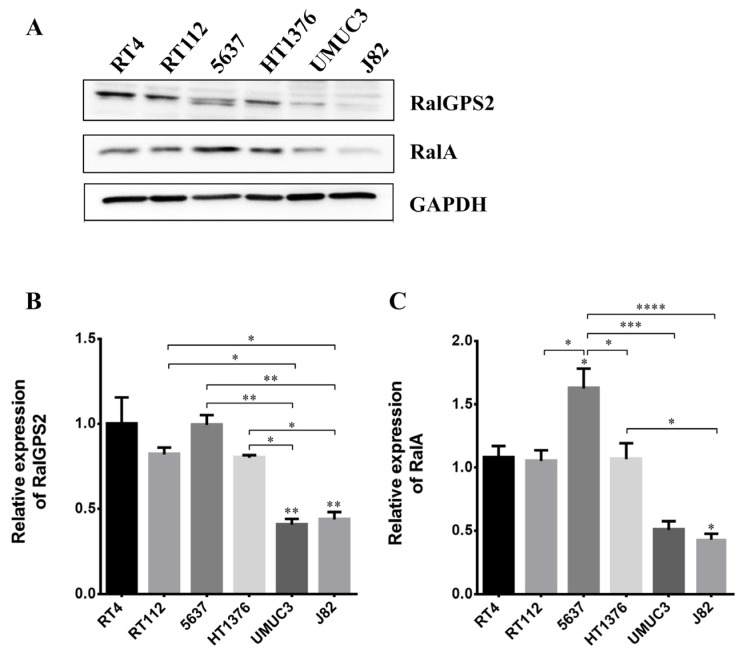
RalGPS2 and RalA expression in bladder cancer cell lines. RT4, RT112, 5637, HT1376, UMUC-3 and J82 cell lysates were separated on SDS-PAGE and blotted to nitrocellulose membrane; blots were probed with anti-RalA or anti-RalGPS2 or anti-GAPDH antibodies. GAPDH was used as a loading control. Panel (**A**): representative Western blot results are shown. Panel (**B**): histograms relative to the quantification of RalGPS2 bands. Panel (**C**): histograms relative to the quantification of RalA bands. Data are expressed as mean ± S.E.M. from three independent experiments. Differences among groups were analyzed using a one-way analysis of variance (ANOVA) followed by Tukey’s post hoc test. * *p* < 0.05, ** *p* < 0.01, *** *p* < 0.001, **** *p* < 0.0001.

**Figure 6 cancers-13-06330-f006:**
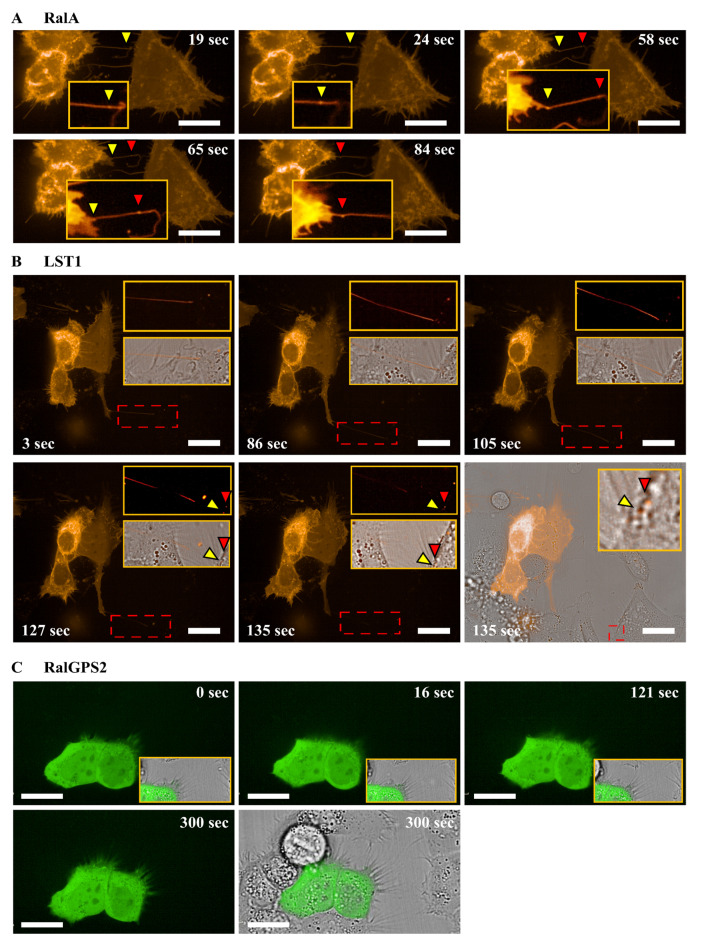
Intracellular transport via TNTs of mobile RalA and LST1 in 5637 cells. 5637 cells were seeded at the concentration of 1 × 10^4^ cells/well on Cell Imaging 24-well Plates. After 24 h, cells were transfected with (**A**) mCherry-RalA or (**B**) mCherry-LST1 or (**C**) pEGFP-RalGPS2; the day after, time-lapse imaging was performed using Operetta CLS™ equipped with 63× immersion objective. Scale bar: 20 µm. Yellow and red triangles indicate transfected proteins transferred from one cell to another one. Magnified views of vesicles containing transfected proteins inside nanotubes are shown.

**Figure 7 cancers-13-06330-f007:**
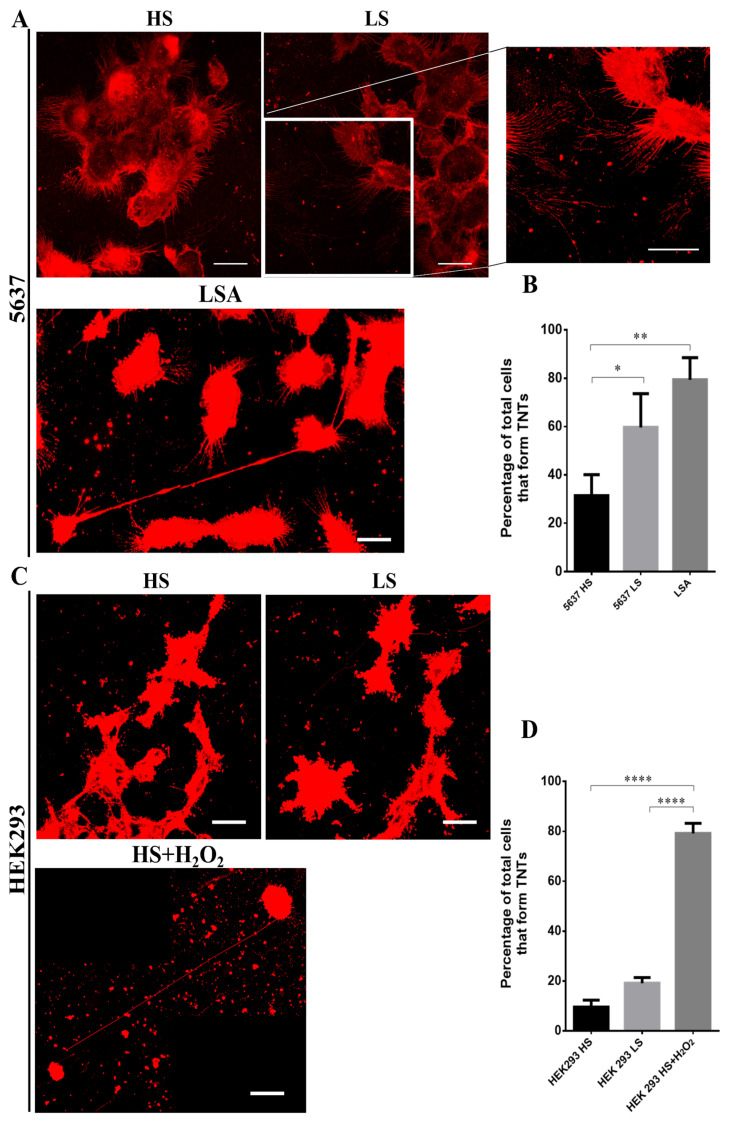
Effect of stress conditions on TNTs formation. (**A**) 5637 and (**C**) HEK293 cells were plated at a density of 1 × 10^5^ cells/mL on porcine gelatin pre-treated coverslips. The next day, cells were placed under several growth conditions as indicated in each panel and, after 24 h, were stained with the membrane dye DiI, fixed, permeabilized and imaged via confocal microscopy. Scale bar: 20 µm. Magnified views of nanotubes are shown. HS (25 mM glucose, 10% FBS, pH 7.4) = High Serum; LS (25 mM glucose, 2.5% FBS, pH 7.4) = Low Serum; LSA (50 mM glucose, 2.5% FBS, pH 6.6) = Low Serum Acidified; HS+H_2_O_2_ (25 mM glucose, 10% FBS, pH 7.4, 100 µM H_2_O_2_). Histograms relative to the percentage of cells forming TNTs in (**B**) 5637 and (**D**) HEK293 cells. At least 200 cells were analyzed per group in three independent experiments. Data are expressed as mean ± S.E.M. from three independent experiments. Differences among groups were analyzed using the one-way analysis of variance (ANOVA) followed by Tukey’s post hoc test. * *p* < 0.05, ** *p* < 0.01, **** *p* < 0.0001.

**Figure 8 cancers-13-06330-f008:**
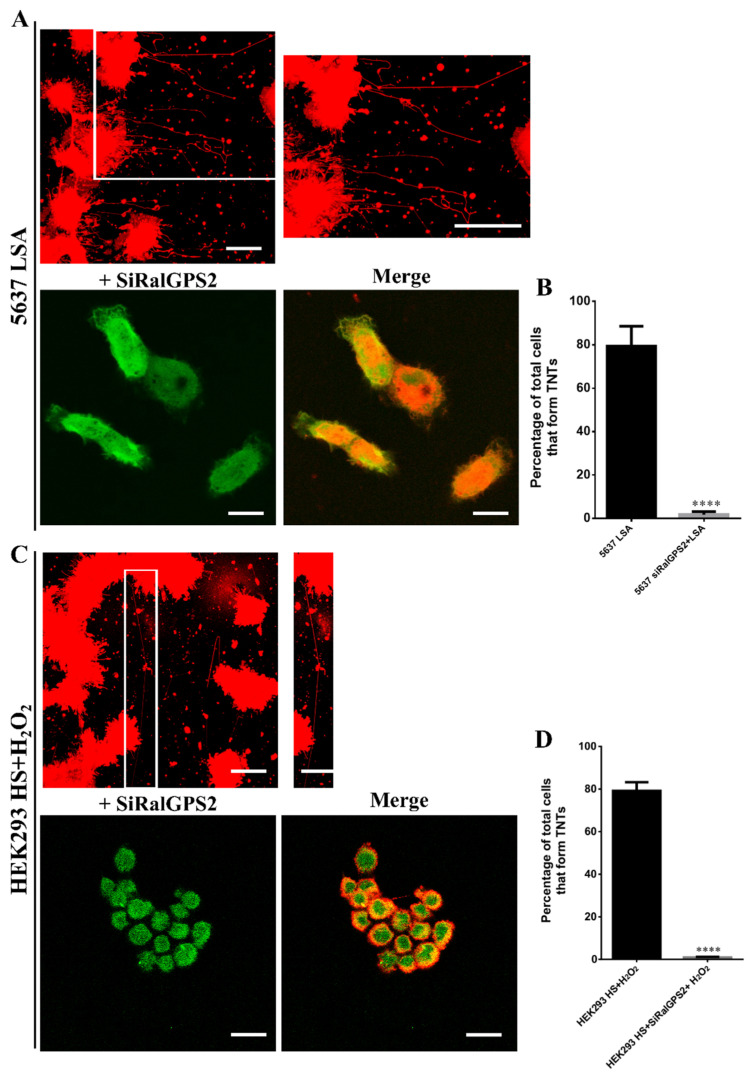
Effect of RalGPS2 depletion under stress conditions on TNTs formation. (**A**) 5637 and (**C**) HEK293 cells were plated at a density of 1 × 10^5^ cells/mL on porcine gelatin pre-treated coverslips. The next day cells were transfected with the specific siRalGPS2 (green), placed under several growth conditions, as indicated in each panel, and, after 24 h, were stained with the membrane dye DiI (red), fixed, permeabilized and imaged via confocal microscopy. Scale bar: 20 µm. Magnified views of nanotubes are shown. LSA (50 mM glucose, 2.5% FBS, pH 6.6) = Low Serum Acidified; HS+H_2_O_2_ (25 mM glucose, 10% FBS, pH 7.4, 100 µM H_2_O_2_). Histograms relative to the percentage of cells forming TNTs in (**B**) 5637 and (**D**) HEK293 cells. At least 200 cells were analyzed per group in three independent experiments. Data are expressed as mean ± S.E.M. from three independent experiments. Differences among groups were tested for significance by Student’s t-test. **** *p* < 0.0001.

**Figure 9 cancers-13-06330-f009:**
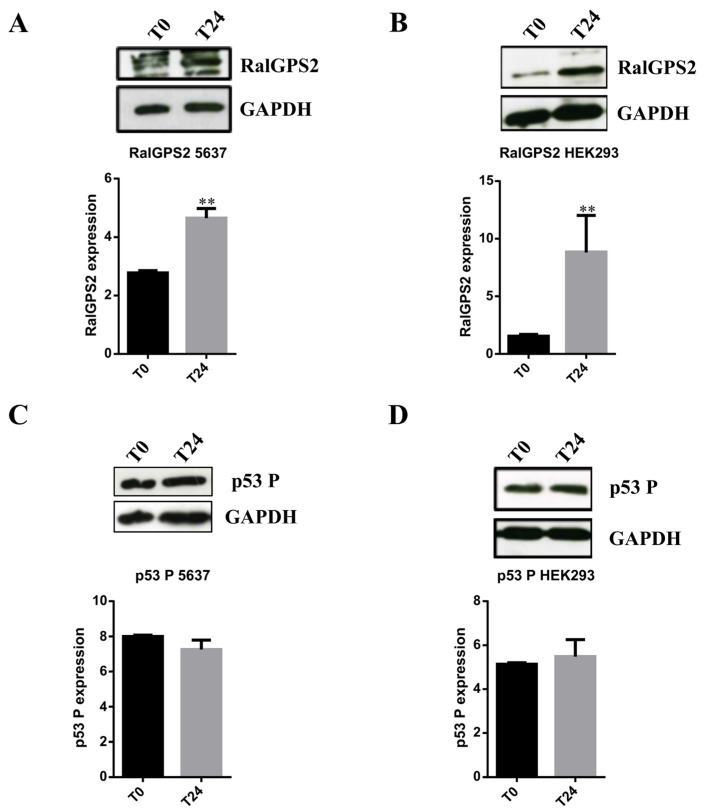
Stress conditions boost RalGPS2 expression in 5637 and HEK293 cells. 5637 and HEK293 cell lysates were separated on SDS-PAGE and blotted to nitrocellulose membrane; blots were probed with anti-RalGPS2 or anti-phospho p53 (p53 P) or anti-GAPDH antibodies. GAPDH was used to normalize sample loading. Panels (**A**,**C**): expression of RalGPS2 and p53 P in 5637 cells treated with LSA medium for 24 h. The panels show representative western blot results and the quantification of (**A**) RalGPS2 or (**C**) p53 P expression. Panels (**B**,**D**): expression of RalGPS2 and p53 P in HEK293 cells treated with HS+H_2_O_2_ medium for 24 h. The panels show representative western blot results and the graphical representation of (**B**) RalGPS2 or (**D**) p53 P expression. Data are expressed as mean ± S.E.M. from three independent experiments. Differences among groups were analyzed using Student’s *t*-test. ** *p* < 0.01.

**Figure 10 cancers-13-06330-f010:**
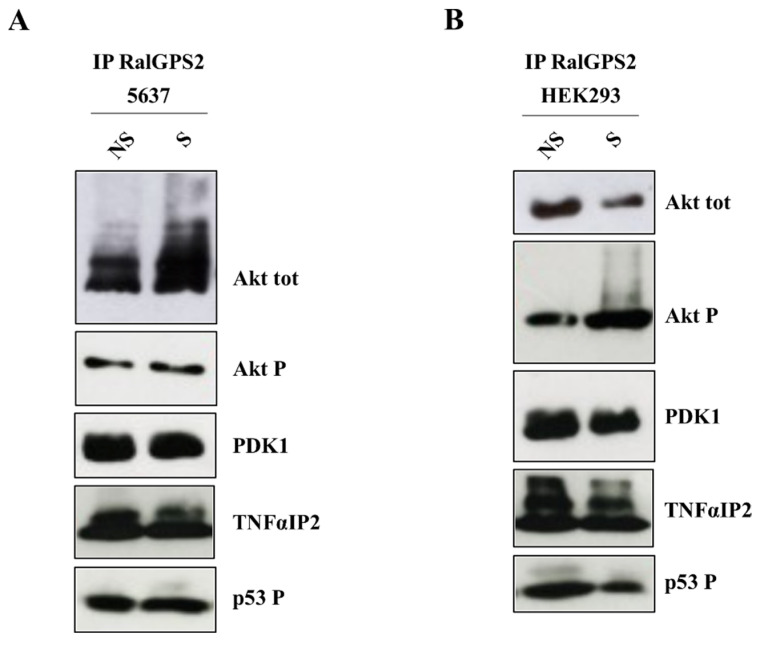
RalGPS2 interacts with Akt, PDK1, TNFαIP2 and p53 P in 5637 and HEK293 cells. (**A**) 5637 and (**B**) HEK293 cells were plated in 100 mm dishes and the day after were first serum-starved for 18 h and then stimulated with 10% FBS for 15 min (S) or left unstimulated (NS). Subsequently, immunoprecipitation (IP) with anti-RalGPS2 antibodies was performed, as indicated in each panel. Immunoprecipitates were assessed for the presence of total Akt (Akt tot), phospho-Akt (Akt P), PDK1, TNFαIP2 and phospho-p53 (p53 P). Three independent experiments were performed.

**Figure 11 cancers-13-06330-f011:**
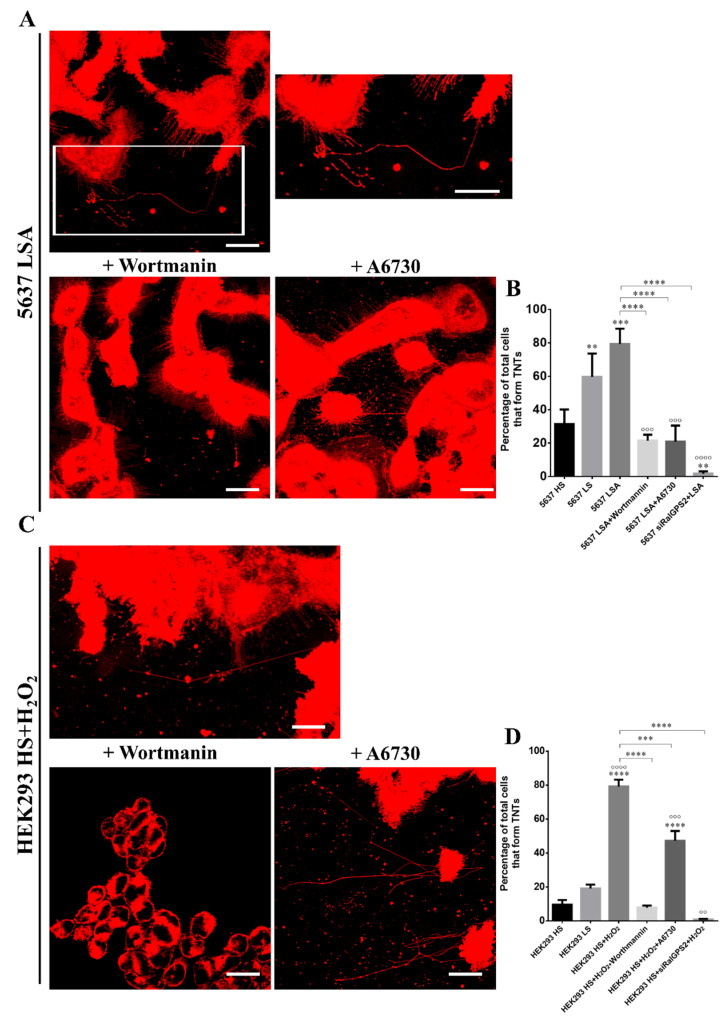
Effect of PI3K and Akt inhibition under stress conditions on TNTs formation. (**A**) 5637 and (**C**) HEK293 cells were plated at a density of 1 × 10^5^ cells/mL on porcine gelatin pre-treated coverslips. The next day the medium was replaced with LSA or HS+H_2_O_2_, respectively, and 24 h later, the cells were treated with or without 100 nM Wortmannin or 10 µM A6730, as indicated in each panel. After 1h of treatment, cells were stained with the membrane dye DiI, fixed, permeabilized and imaged via confocal microscopy. Scale bar: 20 µm. Magnified views of nanotubes are shown. Histograms relative to the percentage of cells forming TNTs in (**B**) 5637 and (**D**) HEK293 cells. At least 200 cells were analyzed per group in three independent experiments. Data are expressed as mean ± S.E.M. from three independent experiments. Differences among groups were tested for significance by the one-way analysis of variance (ANOVA) followed by Tukey’s post hoc test. ** *p* < 0.01, *** *p* < 0.001, **** *p* < 0.0001. °° *p* < 0.01, °°° *p* < 0.001, °°°° *p* < 0.0001 significance compared to LS. HS (25 mM glucose, 10% FBS, pH 7.4) = High Serum; LS (25 mM glucose, 2.5% FBS, pH 7.4) = Low Serum; LSA (50 mM glucose, 2.5% FBS, pH 6.6) = Low Serum Acidified; HS+H_2_O_2_ (25 mM glucose, 10% FBS, pH 7.4, 100 µM H_2_O_2_).

**Figure 12 cancers-13-06330-f012:**
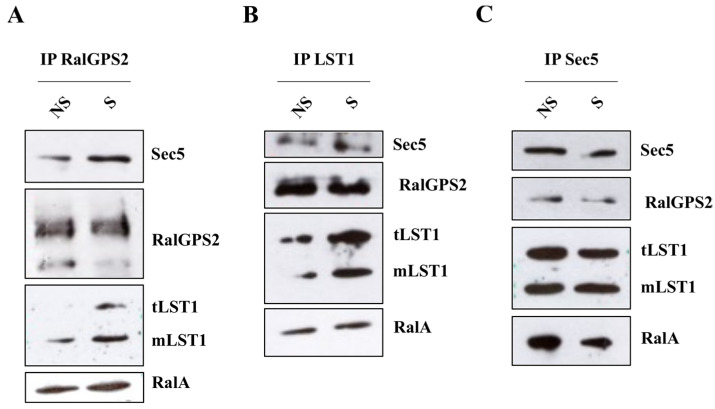
RalGPS2 interacts with RalA, LST1 and Sec5 in HEK293 cells. HEK293 cells were plated in 100 mm dishes, and the day after, were first serum-starved for 18 h and then stimulated with 10% FBS for 15 min (S) or left unstimulated (NS). Subsequently, immunoprecipitation (IP) with (**A**) anti-RalGPS2, (**B**) anti-LST1 or (**C**) anti-Sec5 antibodies were performed, as indicated in each panel. Immunoprecipitates were probed for Sec5, LST1, RalA and RalGPS2 presence. tLST1: LST1 trimerous; mLST1: LST1 monomer.

## Data Availability

The data presented in the current study are available from the first and corresponding author (M.C.) upon reasonable request.

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
