# Peer review of "RalGPS2 Interacts with Akt and PDK1 Promoting Tunneling Nanotubes Formation in Bladder Cancer and Kidney Cells Microenvironment"

_cancers, 2021, doi:10.3390/cancers13246330_

Round 1
Reviewer 1 Report
Comments
Based on previous findings on a complex including RalGPS2 and RalA which is involved in formation of tunneling nanotubes (TNTs) in the bladder carcinoma cell line 5637, the authors extend their knowledge in the same cell line and others. They show that TNTs are built by higher stage bladder cancer cell lines enabling them to exchange mitochondria and proteins and that RalGPS2 interacting with Akt and PDK1 is essential for TNT formation. These are interesting findings shedding light on molecular mechanisms behind cancer cell interactions among each other.
Minor points
Quantification of TNTs: How were the TNTs calculated? Cells counted by eye? Just percentage estimated? Did you count any cell with at least one TNT or was there a threshold? Please explain.
Just for interest: Did you observe some exchange in cells like RT-112 also, perhaps using very short nanotubes?
Is the formation of TNTs dependent on cell density? Please include this issue.
Fig. S3: Legend is missing, please add. Especially 5637 with LS+H2O2 seem to have more dying cells although they grow, what did you observe?
Fig. 7: HEK293 cells with HS and LS look slightly different, what about the p value? 5637 cells in LS look like more than 50% TNT formation, sure about the 50%?
Fig. 8: Please explain which is red, which is green. Cell names at the left would be nice (for Fig. 11, too).
Activation of p53 could be time-dependent, did you look at that? P53 expression should be added, because p53 levels could change. The statement that the amount of active p53 didn´t change should be handled with care.
Figure legends should explain T0 and T24 or use 0 and 24 h, also T3. p53 P should be explained.
3.7: Ref. 40 should be placed after “Akt activation in 5637 cells”.
Fig. 10: How many experiments were done? Akt tot should be explained.
Fig. 12: tLST1 and mLST1 should be explained.
Discussion: “by Rustom” Please add “et al.”.
Please remove green bar before “neuronal” and “e” in “panel of e six cell lines”.
In the current work you didn´t show that blockage of PI3K activity inhibited the production of PIP2 and PIP3 and that this is the cause for less RalGPS2 at the membrane, but you can assume. Please adjust.
Figure WB is missing, please add.
Fig. S1: Please describe abbreviation DPC and brightfield (left), DPC (right).
Grammar/spelling/consistent form: Please correct, e.g. first cited reference and reference list, singular/plural like cell(s) or with verbs, exponent within 1x 104 or 105 cells/well, figura…
Abbreviations and locations of companies: Please (just) explain when first mentioned. Please correct ATCC explanation. Please explain nr.
Author Response
Dear Reviewer,
Thanks for your comments. Here you will find our response to your reviews, point by point; all the questions you asked are highlighted in light blue while the answers are not highlighted.
Minor points
-Quantification of TNTs:
- How were the TNTs calculated?
The TNTs were calculated as the percentage of cells able to display tunneling nanotubes.
- Cells counted by eye?
Cells were counted by eye
- Just a percentage estimated?
Yes, we have just evaluated the percentage of cell forming TNTs
- Did you count any cell with at least one TNT or was there a threshold?Please explain.
We count any cell with at least one TNT. We have added this information in paragraph 2.4.
-Just for interest: Did you observe some exchange in cells like RT-112 also, perhaps using very short nanotubes?
We haven’t observed the formation of nanotubes in RT112; for this reason we haven’t analyzed transfer in this cell line.
-Is the formation of TNTs dependent on cell density? Please include this issue.
Cell density determines the distance and proximity between cells and consequently influences the probability for cells not only to form TNTs but also to form different categories of TNTs ( M. Bénard, 2015 Biology of the Cell,DOI: 10.1111/boc.201500004).
Bernard M. et at have demonstrated in PC12 cells that when cell density is low TNTs are long several tens of micrometers with a diameter of 100-650 nm and contain both actin and tubulin; when cell density increases, there is a higher number of TNTs containing only actin that presents length of max 20 μm with a diameter of 70-200 nm.
We have added this information in the first chapter of the results (3.1).
-Fig. S3: Legend is missing, please add. Fig.S3 has been added
-Especially 5637 with LS+H2O2 seem to have more dying cells although they grow, what did you observe?
Inadequate nutrient intake leads to oxidative stress disrupting homeostasis, activating signaling, and altering metabolism.
What we observe from the experiments conducted is that a low concentration of serum increases the formation of TNTs in 5637 cells; at the same time, it is also true that the use of H2O2 causes suffering in cells that stop making nanotubes but in part continue to grow.
White EZ (White EZ et al. 2020 SCi. Rep. DOI: 10.1038/s41598-020-68668-x) demonstrated in prostate cancer cells that reducing serum (starved) induced reactive oxygen species which provided an early oxidative stress environment and allowed cells to confer adaptability to increased oxidative stress (H2O2). Measurement of cell viability demonstrated a low death profile in stressed cells (starved + H2O2), while cell proliferation was stagnant. Stressed cells also presented a quiescent phenotype suggesting a mechanism of tolerance.
So our cells probably undergos to the same mechanism of adaptation.
-Fig. 7: HEK293 cells with HS and LS look slightly different, what about the p value? 5637 cells in LS look like more than 50% TNT formation, sure about the 50%?
The p value is 0.1153. As we can observe by Fig.7 B the percentage of 5637 cells displaying TNT in LS is 59.6%, we are sure about this result because the statistical analysis was performed on a huge number of cells.
-Fig. 8: Please explain which is red, which is green.
This information has been added in the legend of the figure; green is the specific siRNA for RalGPS2 while red is the membrane dye DiI
-Cell names at the left would be nice (for Fig. 11, too).
We have placed cell names at the left in figure 8 and 11.
-Activation of p53 could be time-dependent, did you look at that?
We have initially analyzed p53 activation at short times (5 min, 15 min, 30 min, 45 min, 60 min, 180 min) but there wasn’t an increase in p-53 phosphorylation both in 5637 and HEK293 cells (data not shown). So, we performed the same analyzes at the same time point in which the RalGPS2 expression increases, namely 24h.
-P53 expression should be added, because p53 levels could change. The statement that the amount of active p53 didn't change should be handled with care.
We agree with your comments; p53 role should be analyzed better because, even if its level doesn’t seem to increase, it interacts with RalGPS2 and so probably it could have some important role in the complex. Furthermore, in future , it would be interesting also to perform an immunoprecipitation assay of RalGPS2 on cells grown in stress condition to analyze possible variation of p53-P. This could tell us that although it seems that active p53 does not vary in response to stress,there could be a pool of active p53 that binds more/better to the complex leading to the formation of TNTs. In response to stress p53 increases the levels of EGFR which in turn determines the activation of the PI3K pathway and consequently leads to the formation of TNTs (Wang Y et al. 2011 Cell Death Differ. doi: 10.1038/cdd.2010.147.) In this paper we have shown that PI3K is also responsible for the localization of RalGPS2 in the membrane (PI3K produces PIP2 and PIP3) and therefore for its activation resulting in the formation of TNTs. Furthermore, p53-P also activates m-sec which interacts with RalGPS2 (Fig.12).
-Figure legends should explain T0 and T24 or use 0 and 24 h, also T3.
We have added the timing to the legend.
- p53 P should be explained.
p53 P has been explained in figure legends
-3.7: Ref. 40 should be placed after “Akt activation in 5637 cells”.
Ref. 40 was added after “Akt activation in 5637 cells”.
-Fig. 10: How many experiments were done?
We have done three independent experiments. We have added this information into Fig.10 legend.
-Akt tot should be explained.
Akt tot has been explained in figure legends
-Fig. 12: tLST1 and mLST1 should be explained.
We have added it (tLST1: LST1 trimerous; mLST1: LST1 monomer).
-Discussion: “by Rustom” Please add “et al.”.
We have added it
-Please remove the green bar before “neuronal” and “e” in “panel of e six cell lines”.
We have removed it and corrected it.
-In the current work you didn't show that blockage of PI3K activity inhibited the production of PIP2 and PIP3 and that this is the cause for less RalGPS2 at the membrane, but you can assume. Please adjust.
This assumption is just present in the discussion: “RalGPS2, as described by Ceriani et al. 2007 is a Ras-independent GEF for RalA containing Cdc25-like GEF domain. The main characteristic of RalGPS2 contains a is its pleckstrin homology (PH) domain, in the present at the C terminus., The latter that binds is able to bind preferentially binds phosphatidylinositol-4,5-bisphosphate (PIP2) but, at a lower extent, also binds phosphatidylinositol 3-4-5-triphosphate (PIP3), which are both PI3K products”.
-Figure WB is missing, please add.
This was an error; we have added to supplementary a file containing all the blots, as the journal requested.
-Fig. S1: Please describe the abbreviation DPC and brightfield (left), DPC (right).
We have described the abbreviations.
-Grammar/spelling/consistent form: Please correct, e.g. first cited reference and reference list, singular/plural like cell(s) or with verbs, exponent within 1x 104 or 105 cells/well, figura…
We have done all the corrections requested.
-Abbreviations and locations of companies: Please (just) explain when first mentioned.
In this respect we will take into account the editor's suggestions.
-Please correct ATCC explanation.
We have corrected with American Type of Culture Collection
-Please explain nr.
nr stays for not-reported/unknown. We have added this information in the text.
Reviewer 2 Report
Dear Authors,
Please add all original western blots to the supplementary material as well.
RalGPS2 interacts with Akt and PDK1, promoting Tunneling nanotubes formation in bladder cancer and kidney cells microenvironment
The authors provide an extensive amount of work demonstrating the involvement of RalA and RalGPS2 in the formation of tunneling nanotubes (TNT). It is interesting how the authors relate to the molecular machinery forming the tunneling nanotubes and the transport of organelles and other components through the nanotubes. There are some remaining ideas and concerns that readers might have difficulties understanding if RalA and RalGPS are necessary for tube formation or transport within the tube. Maybe the authors could even consider adding experiments and preparing alone standing manuscripts focusing on the molecular machinery and transport. In addition, there are some other thoughts listed in the comments below.
Major comments:
Would it be possible to add experiments demonstrating the functionality of the mitochondria after the transfer?
In general, UMUC3 and J82 have approx. 5 and 10% fewer TNTs compared to 5637; however, the expression of RalA is downregulated compared to 5637. Would the authors suggest another molecular mechanism? The authors demonstrate that cells that do not form TNTs have the same expression of RalGPS2. What does that indicate?
In the discussion, the authors write about oxidative stress. Would it be feasible to add experiments in hypoxic conditions? What about the functionality of the mitochondria after stress induction and in relation to transfer?
Would the authors suggest that RalA travels through the TNTs alone, complexed or in a vesicle?
To complete the statement that RalA and RalGPS are needed for TNT formation and other proteins, the authors may consider adding western blot experiments evaluating the expression of the proteins of the TNT machinery (i.e. myosin x, M-sec, etc.).
The authors state that p53 is crucial for TNT formation, however in figure 9 it is not altered after stress induction. What does it mean?
Would the authors consider adding data on the transfer of components between different cell types?
In cancer cells, the signaling via the Akt-PI3K is usually altered and overactive. Would the assumption be correct that the role of the Akt-PI3K pathway only relates to cancer?
Minor comments:
It would be appreciated if the authors could clarify how the 200 cells were analyzed for the presence of TNTs as described in 2.4. Did the authors consider different images (view fields), or were 200 cells taken from one view field. Was the analysis done at the same magnification given in the legends or a smaller one?
Did the authors quantify the mitochondria located in the tubes during transfer?
What are the general mitochondria count per cell line? Some appear to have more mitochondria than other cell lines (UMUC3 > 5637).
In the acidic medium, the glucose content is higher. Could that have an impact on the formation of TNTs as well?
How was the medium acidified?
In principle, the high serum medium is the standard medium. Authors may consider rephrasing in the manuscript that the increased serum medium is the culture medium and, therefore, control.
The authors may directly add to the merged images in figure 8, which protein the green and red color refer to.
Is it possible to add data on siRNA knock-down in the normal medium?
Best regards
Author Response
Dear Reviewer,
Thanks for your comments. Here you will find our response to your reviews, point by point; all the questions you asked are highlighted in light blue while the answers are not highlighted.
-The authors provide an extensive amount of work demonstrating the involvement of RalA and RalGPS2 in the formation of tunneling nanotubes (TNT). It is interesting how the authors relate to the molecular machinery forming the tunneling nanotubes and the transport of organelles and other components through the nanotubes. There are some remaining ideas and concerns that readers might have difficulties understanding if RalA and RalGPS are necessary for tube formation or transport within the tube. Maybe the authors could even consider adding experiments and preparing alone standing manuscripts focusing on the molecular machinery and transport. In addition, there are some other thoughts listed in the comments below.
RalA GTPase importance in TNTs formation has been reported in numeros papers and reviews.
Hase K et al. (Hase K. et al. 2009 Nat Cell Biol. DOI: 10.1038/ncb1990) have demonstrated that M-Sec promotes membrane nanotube formation by interacting with RalA and the exocyst complex and that RNAi-mediated depletion of RalA or the exocyst subunits, Sec5 or Sec6, significantly impairs M-Sec-induced membrane extension.
Furthermore overexpression of mutant RalA48W or RalA38R that cannot bind to the exocyst subunits, Exo84 or Sec5 respectively, significantly reduces M-Sec-induced TNT formation (Schiller C. J Cell Sci. 2013 DOI: 10.1242/jcs.114033).
Concerning RalGPS2 we have demonstrated in this paper and in D’Aloia et al. 2018 (D’Aloia et al. 2018 Exp Cell Res DOI: 10.1016/j.yexcr.2017.11.036) that it is involved in TNTs formation; specific siRNA for RalGPS2 impairs TNTs formation. RalGPS2 is not transported within the tube and there is no evidence concerning its involvement in the transport along the tube.
So, both proteins are implicated in TNTs formation and there is no data about their involvement in transport within the nanotube.
Thanks for your comment; we will take into account the idea to explore the role of the complex in transport in a future work.
Major comments:
-Please add all original western blots to the supplementary material as well.
We have uploaded in the specific section and in Supplemental materials a file named “Uncropped western blot” in which there are the original uncropped whole western blots showing all the bands with all molecular weight markers on the Western.
In this respect, I would like to specify that filters have been cut to see more different proteins in the same gel or in different gels. Furthermore, if not specified, the markers present are only those that include the range of molecular weights in which our protein is located(as indicated in figures inside the manuscript); if the below or above area of the blot is not shown, it is because that area has been used to visualize other proteins.
-Would it be possible to add experiments demonstrating the functionality of the mitochondria after the transfer?
Our main goal in this manuscript was to understand whether the nanotubes we observed were functional or not. To address this question we test mitochondria transfer, thus among organelles that can be exchanged by TNTs, they are those more transferred in cancer cells responsible for resistance mechanisms. Therefore, the purpose was not, in this article, to perform a metabolic analysis on these cells. These analyses will be considered for future projects and will be performed using heterotipic cultures with cells with different metabolic properties as described in a recent paper published in Nature Nanotechnology (Saha T. et al. 2021 Nature Nanotechnology DOI: 10.1038/s41565-021-01000-4).
However, to partially answer your question, we verified (Supplementary figure S2 ) if the transferred mitochondria were functional using a double staining of mitochondria: MitoTracker™ Green FM and Tetramethylrhodamine, ethyl ester (TMRE). The first is a green-fluorescent mitochondrial dye which localize to mitochondria regardless of mitochondrial membrane potential and the second one is a cell-permeant, cationic, red-orange fluorescent dye that is readily sequestered by active mitochondria and accumulates in the negatively charged mitochondrial matrix according to the Neust equation potential, thus correlating with mitochondrial membrane potential. So MitoTracker™ Green allows to see all mitochondria while TMRE stains active mitochondria.
As shown in Supplementary figure S2 there is no correlation between mitochondrial trafficking and functionality during transport. Bladder cancer cells are able to exchange both functional and non-functional mitochondria.
In general, UMUC3 and J82 have approx. 5 and 10% fewer TNTs compared to 5637; however, the expression of RalA is downregulated compared to 5637. Would the authors suggest another molecular mechanism?
UMUC3 cells express an oncogenic hyperactivated mutant of KRas (KRasG12C). So, pathways downstream to Ras, as RalGDS (Ral GEFs Ras-dependent) and Akt, are activated leading to TNTs formation.
In J82 cells, although there is a low level of Ral A expression, the protein is predominantly in the GTP-bound active state (Smith SC 2007Clin. Cancer Res. 2007, https://doi.org/10.1158/1078-0432.CCR-06-2419.).
The authors demonstrate that cells that do not form TNTs have the same expression of RalGPS2. What does that indicate?
Here we have shown that there is no correlation between the expression levels of RalA and RalGPS2 with the tunneling nanotubes formation. Here and in a previous article (D’Aloia A. et al. 2018 Exp Cell Res. DOI: 10.1016/j.yexcr.2017.11.036) we have shown instead that the formation of nanotubes depends on the function of these proteins. In fact, their knockdown compromises the ability to form nanotubes even under stressful conditions. A cellular function is often not related to the expression levels of a protein, but to its activation state.
Regarding RalGPS2, it is a poorly characterized multi-domain functional protein (classified as Tdark by the Illuminating the Druggable Genome Knowledge Management Center (IDG-KMC); https://pharos.nih.gov/), whose activity is likely finely regulated in a stimulus-dependent manner by inter- and / or intra-molecular interactions and subcellular translocations, similarly to most of the signal transduction proteins in the Ras superfamily pathways, as Ras- and Ral-specific GEFs (Sacco E, Farina M, Greco C, Lamperti S, Busti S, Degioia L, Alberghina L, Liberati D, Vanoni M. The regulation of hSos1 activity is a system-wide property generated by its multi-domain structure Biotechnol Adv. 2012 Jan-Feb; 30 (1): 154- 68.doi: 10.1016 / j.biotechadv.2011.07.017. Epub 2011 Aug 6. PMID: 21851854; Tian X 2002 Embo J. DOI: 10.1093/emboj/21.6.1327). These complex mechanisms of activation of RalGPS2 will be further investigated in future studies.
-In the discussion, the authors write about oxidative stress.
Would it be feasible to add experiments in hypoxic conditions?
Yes, it would be interesting to perform stress tests under hypoxia conditions, which could be chemically induced by cobalt chloride as reported by Munoz-Sanchez J. (Monuz-Sanchez J. and Chanez-Cardenas M.E. 2019 J. Appl. Toxicol. DOI: 10.1002/jat.3749). This aspect will be examined in depth in a future work focused on metabolic aspects related to nanotubes formation and trafficking.
What about the functionality of the mitochondria after stress induction and in relation to transfer?
To answer your question we attempted to perform a mitochondria transfer experiment in stress media. Unfortunately, cells grown for 24 h in stress media die after MitoTracker™ Green FM and TMRE staining; it is well known that these dyes are toxic for cells.
These analyses will be considered for future projects and will be performed using heterotipic cultures with cells with different metabolic properties as described in a recent paper published in Nature Nanotechnology (Saha T. et al. 2021 Nature Nanotechnology DOI: 10.1038/s41565-021-01000-4).
Would the authors suggest that RalA travels through the TNTs alone, complexed or in a vesicle?
This aspect has not been evaluated in the present study. The only studies in which a GTPase is transferred by TNTs are that of Rainy N. (Rainy N. et al. 2013 Cell Death Dis. DOI: 10.1038/cddis.2013.245) and that of Desir S. (Desir S. et al 2019 Cancers DOI: 10.3390/cancers11070892). Rainy has demonstrated that the GTPase H-Ras is transferred from B to T cells via tunneling nanotubes and that H-Ras is transferred with membrane patches containing it. Desir S. shows TNTs mediate intercellular transfer of mutant KRAS in recipient colon cancer cells and that mutant KRAS is transferred from more aggressive to less aggressive CRC cells.
We could speculate that since RalA is anchored to the plasma membrane, thanks to its post-translational modification at its C-Terminal, it probably is transferred as membrane patches containing it as for H-Ras.
-To complete the statement that RalA and RalGPS are needed for TNT formation and other proteins, the authors may consider adding western blot experiments evaluating the expression of the proteins of the TNT machinery (i.e. myosin x, M-sec, etc.). To partially answer your question we have evaluated M-Sec, alias TNF⍺IP2 presence in immunoprecipitation complexes both in 5637 and in HEK293 cells. As it is shown in figure 10, TNF⍺IP2 co-immunuprecipitates with RalGPS2 in both cell lines. M-Sec likely coordinates with RalA and the exocyst complex to initiate TNTs formation (Ohno H. et al. 2010 Communicative & Integrative Biology DOI: 10.4161/cib.3.3.11242). M-Sec-induced TNTs are associated with F-actin but not with microtubules (Hase K. et al. 2009 Nat Cell Biol DOI: 10.1038/ncb1990).
As regards Myosin, we actually haven’t data because our paper is focused on the Akt/PDK1 pathway. In the future we will analyze other proteins that could be part of the complex under different metabolic perturbations to further characterize the mechanisms underlying the formation of nanotubes.
The authors state that p53 is crucial for TNT formation, however in figure 9 it is not altered after stress induction. What does it mean? We agree with your comments; p53 role should be analyzed better because, even if its level doesn’t seem to increase, it interacts with RalGPS2 and so probably it could have some important role in the complex. So, to better comprehend the p53 role it would be interesting to perform immunoprecipitation assays in stress conditions. Indeed, although it seems that active p53 does not vary in response to stress,there could be a pool of active p53 that binds more/better to the complex that leads to the formation of TNTs.
-Would the authors consider adding data on the transfer of components between different cell types? In this work we focused our attention only on homotypic cultures. The idea of analyzing the formation of nanotubes and the transport of organelles, such as mitochondria, between different cell types is certainly one of the goals of our future research.
-In cancer cells, the signaling via the Akt-PI3K is usually altered and overactive. Would the assumption be correct that the role of the Akt-PI3K pathway only relates to cancer? As all known signaling pathways including the Akt/PI3K axis have physiological roles in development and disease (Dummler B. et al 2007 Biochem Soc Trans DOI: 10.1042/BST0350231).
Elevated PI3K signaling is considered a hallmark of cancer in fact many PI3K pathway-targeted therapies have been tested in oncology trials. Investigations in other fields have uncovered exciting and often unpredicted roles for PI3K catalytic and regulatory subunits in normal cell function and in disease. For example too little PI3K signaling in the liver and muscle can lead to insulin resistance and type 2 diabetes. Class I PI3K signaling is activated by antigen receptors expressed by T and B cells, and by other inputs including costimulatory molecules and cytokine receptors. Several comprehensive reviews have detailed how PI3K signaling is engaged by different receptors to regulate a variety of lymphocyte responses, and how genetic deficiency or hyperactivity of PI3K isoforms can both lead to immunodeficiency (Hawkins and Stephens, 2015; Lucas et al., 2016; Okkenhaug and Vanhaesebroeck, 2003). There are many non-malignant diseases associated with hyperactive PI3K/mTOR signaling as Cowden Syndrome, CLOVES and other tissue overgrowth syndromes, Tuberous sclerosis and Parkinson’s Disease (Fruman Da et al 2017 Cell doi: 10.1016/j.cell.2017.07.029). So Akt/PI3K is not only associated with cancer.
Minor comments:
-It would be appreciated if the authors could clarify how the 200 cells were analyzed for the presence of TNTs as described in 2.4. Did the authors consider different images (view fields), or were 200 cells taken from one view field. Was the analysis done at the same magnification given in the legends or a smaller one? We use different view fields in the same plate from three different experiments to have truthful and accurate data. The analyses were performed at the same magnification of that reported in figure legends.
-Did the authors quantify the mitochondria located in the tubes during transfer? No, we didn’t for experimental limitations.
What are the general mitochondria count per cell line? Some appear to have more mitochondria than other cell lines (UMUC3 > 5637). The number of mitochondria of the different bladder cancer cell lines under investigation was not reported in the present manuscript because the focus of our work is not the analysis of cellular metabolic functions, but of the Tunneling nanotubes, and the molecular mechanism underlying their formation, and their ability to transport organelles, such as mitochondria. In a previous work we have shown, using the bladder cancer 5637 and RT112 cell lines (Pasquale et al Profiling and Targeting of Energy and Redox Metabolism in Grade 2 Bladder Cancer Cells with Different Invasiveness Properties. Cells. 2020 Dec 11;9(12):2669. doi: 10.3390/cells9122669. PMID: 33322565; PMCID: PMC7764708.), that the mitochondrial mass of a cell line (measured by quantitative imaging on cells stained with the non-potentiometric mitotracker green dye) does not correlate with mitochondrial function (measured with mitotracker Red or TMRE potentiometric dyes or by functional metabolic with Seahorse technology). The analysis of the mitochondrial functions of the cell lines under study is currently ongoing, but preliminary data has highlighted a very complex picture that requires more in-depth analysis to be understood (metabolic functional analysis, and metabolomics). It would therefore be reductive to report information on the number of mitochondria of the different cell lines in this work.
-In the acidic medium, the glucose content is higher. Could that have an impact on the formation of TNTs as well? TNT formation might be regulated in vivo, by nutrient supply, infection, or therapy. The role of glucose is somewhat controversial; in fact there are many papers that provide evidence that both high-glucose and low-glucose could stimulate TNTs formation. For example in vitro, low-serum (2.5% FBS) and high-glucose concentrations (50 mM) were found to stimulate TNT formation between murine K7M2 osteosarcoma cells and MC3T3 osteoblast cells (Thayanithy V. et al 2014 Translational Research.doi: 10.1016/j.trsl.2014.05.011); low-serum, hyperglycemic, acidic growth medium stimulate both the formation of TNTs and the mitochondrial trafficking between malignant or between normal mesothelial cells (Lou E 2012 PLoS One doi: 10.1371/journal.pone.0033093). Furthermore high-glucose concentrations were shown to diminish mitochondrial motility, by a mechanism involving Milton and its O-GlcNAcylation by the O-GlcNAc transferase (OGT) (Pekkurnaz G 2014Cell doi: 10.1016/j.cell.2014.06.007). Albeit the effects of high-glucose concentrations reported above, it is also worth noting that, in other cell systems, it is the glucose deprivation that was found to enhance the TNT-mediated mitochondrial transfer, as observed from MSCs to endothelial cells (Liu K.Microvascular Research. 2014 doi: 10.1016/j.mvr.2014.01.008).
In our work we used the same conditions described in Lou et al. 2012 (Lou E 2012 PLoS One doi: 10.1371/journal.pone.0033093), and Thayanithy V. et al 2014 (Thayanithy V. et al 2014 Translational Research.doi: 10.1016/j.trsl.2014.05.011).
So actually we have no evidence that could let us draw any conclusion about the role of high glucose under acidification condition in TNTs formation in our assays.
-How was the medium acidified? We used a low-buffered medium and we adjust the pH at 6.6
-In principle, the high serum medium is the standard medium. Authors may consider rephrasing in the manuscript that the increased serum medium is the culture medium and, therefore, control. As you suggested we have specified in par. 2.10 that High Serum (HS) medium is the standard culture medium.
-The authors may directly add to the merged images in figure 8, which protein the green and red color refer to. Done; green is the specific siRNA for RalGPS2 while red is the membrane dye DiI.
-Is it possible to add data on siRNA knock-down in the normal medium? Data on siRNA knock-down of RalGPS2 in the normal medium has been previously published in the paper:
D’Aloia, A.; Berruti, G.; Costa, B.; Schiller, C.; Ambrosini, R.; Pastori, V.; Martegani, E.; Ceriani, M. RalGPS2 Is Involved in Tunneling Nanotubes Formation in 5637 Bladder Cancer Cells. Exp. Cell Res. 2018, 362 (2), 349–361.
In this paper it has been demonstrated that RalGPS2 silencing in 5637 grown in normal medium markedly reduced the formation of TNT-like protrusions.
Reviewer 3 Report
RalGPS2 interacts with Akt and PDK1 promoting Tunneling nanotubes formation in bladder cancer and kidney cells microenvironment
Major comments:
The paper analyses tunneling nanotubes (TNT) in bladder cancer cell lines for various stress conditions.
In general, the results of the paper are technically clearly presented with all needed control. The conclusion are supported by the data and the methodology is clearly described. The authors studied TNT on a panel of bladder cancer cells lines with different metastatic potential. Previous studies have shown that TNT formation is induced by different stressors like nutrient deprivation, low pH, hypoxia and oxidative stress and In the present study the authors demonstrate effect of low serum, acidification and oxidative stress on bladder cancer cell lines.
The paper should be interesting to the researches in field of tunnelling nanotubes and cancers cells intercommunication.
However, I have some general comments:
The introduction is too long. Specifically, first paragraphs focus on the problem of high recurrence of bladder cancer, but this part could be significantly shortened.
Also in the seventh paragraph, there is a very long overview of what TNTs are and the historical background. This part also could be shortened without loss of important reference and background.
In the discussion
One of the relevance of the presented study is that it analyses TNT in various bladder cancer cell lines and shows that the more malignant cell lines also have more TNTs.
Are there any in vivo data studying inhibitors that would prevent TNT formation in order to decrease metastatic potential?
The authors show effects of wortmannin, an irreversible PI3K inhibitor which disrupts TNTs development in 5637 and HEK293 cell lines pre-treated with stress media. Are there any in vivo studies of wortmannin or specific in vitro test that would show reduced ability of survival, anchorage-independent growth or migration?
As RalGPS2 silencing caused a strong reduction in the percentage of cells producing TNTs in vitro – are there any in vivo data, which would further support relevance of these observations?
The authors show interesting result that RalA GTPase was transported from cell to cell via TNTs.
As RalA is necessary for the anchorage-independent, did authors check if the anchorage-independent growth is indeed dependent on RalA GTPase for one of their cancer cell lines – e.g. by culturing cells on poly-hema coated plates?
Minor comments:
Conclusions: there is a repeated sentence.
Author Response
Dear Reviewer,
Thanks for your comments. Here you will find our response to your reviews, point by point; all the questions you asked are highlighted in light blue while the answers are not highlighted.
Major comments:
-The introduction is too long. Specifically, first paragraphs focus on the problem of high recurrence of bladder cancer, but this part could be significantly shortened. We have significantly shortened the introduction.
-Also in the seventh paragraph, there is a very long overview of what TNTs are and the historical background. This part also could be shortened without loss of important reference and background. We have shortened the TNTs paragraph.
In the discussion
One of the relevance of the presented study is that it analyses TNT in various bladder cancer cell lines and shows that the more malignant cell lines also have more TNTs.
-Are there any in vivo data studying inhibitors that would prevent TNT formation in order to decrease metastatic potential? There are few studies of TNTs in primary cell cultures derived from patients. In these studies two known inhibitors of TNTs were used: vincristine (VCR) and Cytochalasin D; the first is a inhibitor of microtubule formation while the second causes the disruption of actin filaments and inhibition of actin polymerization. Vincristine was used in an in vivo study recently published by Burt R. et al. (Burt R. et al 2019 Blood doi: 10.1182/blood.2019001398) focused on the mesenchymal stromal cell (MSC) niche in adult acute lymphoblastic leukemia (ALL). Authors have analyzed MSCs, directly isolated from the primary bone marrow specimens of patients with ALL, exposed to the reactive oxygen species (ROS)–inducing chemotherapy agents cytarabine (AraC) and daunorubicin (DNR) demonstrating that activated MSCs prevented therapy-induced apoptosis and death in ALL targets, via mitochondrial transfer through tunneling nanotubes (TNTs). The “rescue” function of activated MSCs was prevented by reduction of mitochondrial transfer by selective mitochondrial depletion or interference with TNT formation by microtubule inhibitor vincristine (VCR). A similar study was performed by Wang J. et al. 2018 (Wang J. et al 2018 Journal of Hematology & Oncology DOI: 10.1186/s13045-018-0554-z) using Cytochalasin D in MSCs and Jukart T-Hall co-cultured cells. They demonstrated that jurkat cells treated with cytochalasin D had an increased apoptosis rate and decreased cell viability , indicating that blocking mitochondria transfer through TNTs decreased the capacity of MSCs to protect Jurkat cells from drug cytotoxicity.
-The authors show effects of wortmannin, an irreversible PI3K inhibitor which disrupts TNTs development in 5637 and HEK293 cell lines pre-treated with stress media. Are there any in vivo studies of wortmannin or specific in vitro test that would show reduced ability of survival, anchorage-independent growth or migration? Wortmannin (Wtmn) failed clinical translation due to drug-delivery challenges although several preclinical data demonstrated its ability to reduce survival, growth and migration in different cancer cells (Yun J, Lv YG, Yao Q, Wang L, Li YP, Yi J. Wortmannin inhibits proliferation and induces apoptosis of MCF-7 breast cancer cells. Eur J Gynaecol Oncol. 2012;33(4):367-9. PMID: 23091892. Tae-Min Cho, Wun-Jae Kim, Sung-Kwon Moon, AKT signaling is involved in fucoidan-induced inhibition of growth and migration of human bladder cancer cells,Food and Chemical Toxicology, Volume 64, 2014, Pages 344-352 https://doi.org/10.1016/j.fct.2013.12.009; Teranishi F, Takahashi N, Gao N, Akamo Y, Takeyama H, Manabe T, Okamoto T. Phosphoinositide 3-kinase inhibitor (wortmannin) inhibits pancreatic cancer cell motility and migration induced by hyaluronan in vitro and peritoneal metastasis in vivo. Cancer Sci. 2009 Apr;100(4):770-7. doi: 10.1111/j.1349-7006.2009.01084.x. PMID: 19469020.).
Recent paper demonstrates that a nanoparticles (NP) formulation of Wtmn can improve solubility and toxicity of this drug. NP Wtmn was shown to be an effective radiosensitizer in vivo using two murine xenograft models of cancer (Karve S, Werner ME, Sukumar R, et al. Revival of the abandoned therapeutic wortmannin by nanoparticle drug delivery. Proc Natl Acad Sci U S A. 2012;109(21):8230-8235. doi:10.1073/pnas.1120508109).
-As RalGPS2 silencing caused a strong reduction in the percentage of cells producing TNTs in vitro – are there any in vivo data, which would further support relevance of these observations? Actually there is no data in vivo on RalGPS2 and on RalGPS2 silencing.
-As RalA is necessary for the anchorage-independent, did authors check if the anchorage-independent growth is indeed dependent on RalA GTPase for one of their cancer cell lines – e.g. by culturing cells on poly-hema coated plates? We haven’t performed studies on the role of RalA in anchorage independent growth in HEK293 and 5637 cells.
In literature there is no data concerning this aspect on the cell lines considered in this work, although it has been demonstrated that RalA silencing reduces anchorage independent growth and tumorigenesis in different cancer cell lines, including colorectal (Martin TD, Samuel JC, Routh ED, Der CJ, Yeh JJ. Activation and involvement of Ral GTPases in colorectal cancer. Cancer Res. 2011;71(1):206–15. https://doi.org/10.1158/0008-5472.CAN-10-1517) and breast (Thies, K.A., Cole, M.W., Schafer, R.E. et al. The small G-protein RalA promotes progression and metastasis of triple-negative breast cancer. Breast Cancer Res 23, 65 (2021). https://doi.org/10.1186/s13058-021-01438-3).
Minor comments:
-Conclusions: there is a repeated sentence.
The repeated sentence in the Conclusions section was removed.
Round 2
Reviewer 2 Report
I have no further comments and appreciate the corrections and additions given in the revised manuscript as well as the explanations in the letter.
Reviewer 3 Report
the authors have addresed all issues and I have no further comments,